# PROCEEDINGS A

category theory, mathematical physics, quantum computing

2-category, knot theory, quantum computing, 2Hilb, graphical calculus, verification

**Author for correspondence:**
David J. Reutter
e-mail: david.reutter@cs.ox.ac.uk

# Shaded tangles for the design and verification of quantum circuits

David J. Reutter[1] and Jamie Vicary[1,2]

[1]Department of Computer Science, University of Oxford, Oxford, UK
[2]School of Computer Science, University of Birmingham, Birmingham, UK

DJR, 0000-0003-4044-360X

We give a scheme for interpreting shaded tangles as quantum circuits, with the property that if two shaded tangles are ambient isotopic, their corresponding computational effects are identical. We analyse 11 known quantum procedures in this way—including entanglement manipulation, error correction and teleportation—and in each case present a fully topological formal verification, yielding generalized procedures in some cases. We also use our methods to identify two new procedures, for topological state transfer and quantum error correction. Our formalism yields in some cases significant new insight into how the procedures work, including a description of quantum entanglement arising from topological entanglement of strands, and a description of quantum error correction where errors are 'trapped by bubbles' and removed from the shaded tangle.

## 1. Introduction

### (a) Overview

In this paper, we introduce a new knot-based language for designing and verifying quantum circuits. Terms in this language are *shaded tangles*, which look like traditional knot diagrams, possibly involving multiple strings and strings with open ends, and decorated with an alternating two-coloured shading pattern, where adjacent regions have distinct shadings. Examples of shaded tangles are given in figure 2, and the notation is defined formally in §2.

**Figure 1.** Part of the graphical language along with its interpretation in terms of quantum structures. (*a*) Qudit identity, (*b*) qudit preparation, (*c*) 1-qudit gate and (*d*) 2-qudit gate. (Online version in colour.)

**Figure 2.** Shaded tangles giving the circuit and specification for constructing a tripartite GHZ state. (*a*) Circuit and (*b*) specification. (Online version in colour.)

We give a mathematical model in which a shaded tangle is interpreted as a linear map between Hilbert spaces. Since this is the basic mathematical foundation for quantum information, this enables us to interpret our shaded tangles as quantum procedures. Under this interpretation, we read our shaded tangles as quantum circuits, with time flowing from bottom to top, and with individual geometrical features of the diagrams—such as shaded regions, cups and caps and crossings—interpreted as distinct quantum circuit components, such as qudits,[1] qudit preparations, and certain 1- and 2-qudit gates (namely, generalized Hadamard and control-Z gates). See figure 1 for this part of the graphical language. The chosen projection of the shaded tangle into two-dimensional space which we use to draw these images affects the specific interpretation as a quantum circuit, with the height function indicating the order in which gates are applied.

Given two shaded tangles with the same shading pattern on their boundaries, we say they are *isotopic* just when, ignoring shading and considering them as ordinary knotted strings, one can be deformed topologically into the other, in the ordinary sense of ambient isotopy for knots. We show that our semantics is *sound* with respect to this isotopy relation: that is, if two shaded tangles are isotopic, then they are equivalent as quantum circuits, in the sense that the quantum computations they describe have identical underlying linear maps.

This yields a method for the design and verification of quantum procedures. We draw one shaded tangle for the *circuit*, describing the exact steps the quantum computer would perform, and another shaded tangle for the *specification*, describing the intended computational effect.[2] The circuit is then verified simply by showing that the two shaded tangles are isotopic. Since humans have an innate skill for visualizing knot isotopy, at least for knots with relatively few crossings, this verification procedure can often be performed immediately by eye.

We illustrate this idea in figure 2, which illustrates the circuit and specification for constructing a *GHZ state*, an important primitive resource in quantum information. It can be seen by inspection that, ignoring shading, the tangles are isotopic, and hence the procedure is correct; by reference to figure 1, we see that the circuit figure 2*a* involves three qudit preparations, two 1-qudit gates and two 2-qudit gates. We give another example in figure 3 of a tangle proof of an error correction

---

[1]A *qudit* is a *d*-dimensional quantum system; a *qubit* is a qudit for $d = 2$.

[2]We note that our work concerns only the logical structure of quantum circuits, and does not make any claims about how these should be physically implemented.

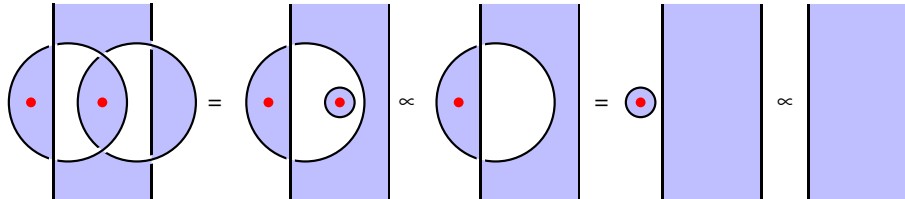

**Figure 3.** Example verification of an error correcting code. (Online version in colour.)

procedure; here the red dots are phase errors, and we see that they can be 'trapped by bubbles' and removed from the composite.

For the remainder of §1, we give an overview of our results, describe their potential significance, highlight some weaknesses of our approach, and give an in-depth analysis of related work. Readers interested only in the mathematical development may skip ahead to §2.

Further discussion and omitted proofs can be found in an electronic supplementary appendix on the website of the Proceedings of the Royal Society.

## (b) Main results

In our main results, we apply this new high-level technique to represent and verify 13 quantum procedures, some generalized from their form in the literature, and some completely new. We give a summary here of the procedures we analyse.

— Section 4a. A generalization of a procedure due to Uchida *et al.* [1] for constructing GHZ states.
— Section 4b. Procedures due to Raussendorf and others [2,3] for constructing *cluster chains* [4], resources of central importance in quantum information.
— Section 4c. Procedures due to Briegel and others [2,5] for interconverting certain GHZ and cluster states.
— Section 4d,e. Procedures due to van den Nest and others [6] for cutting and splicing cluster chains, which play an important role in measurement-based quantum computation [4]. We generalize these procedures to qudits and to new classes of cluster chains.
— Section 4f. A new procedure for robust state transfer in a cluster state-based quantum computer.
— Section 5a. A generalization of procedures due to Karlsson, Hillery, Grudka and others [7–9] for measurement-based teleportation along a GHZ state.
— Section 5b. A generalization of a procedure due to Raussendorf & Briegel [3] for measurement-based teleportation along a cluster chain.
— Section 5c. A procedure for teleportation along an *n*-party GHZ state, which we believe is folklore, with a new robustness property against a broad family of errors.
— Section 5d. A procedure due to Yu, Jaffe and others [10,11] which reduces resource requirements for executing a distributed controlled quantum gate.
— Section 6c,d. The phase code and the Shor code [12–14], important error correcting codes in quantum information, which are built from Hadamard matrices.[3]
— Section 6e. New generalizations of the phase code and Shor code, based on unitary error bases (UEBs)[4] rather than Hadamard matrices.

[3]A *Hadamard matrix* is a unitary matrix with all coefficients having the same absolute value. Hadamard matrices are important primitive structures in quantum information, playing a central role in quantum key distribution and many other phenomena [15].

[4]A *unitary error basis* is a basis of unitary operators on a finite-dimensional Hilbert space, orthogonal with respect to the trace inner product. They provide the basic data for all quantum teleportation and dense coding procedures [16], and some error correction procedures [17,18].

## (c) Significance

We outline some areas of potential significance of our work.

### (i) Novelty

Several of the procedures we verify are in fact generalizations of those described in the literature, or are completely new, thereby making our results potentially of interest in mainstream quantum information science. In particular, we highlight the new procedure for robust state transfer in a cluster-state quantum computer (§4f), and the new constructions of error correcting codes based on UEBs (§6e). In particular, we aim to rebut a common criticism that categorical methods in quantum information are useful only to recast known phenomena in a formal setting, without generating new knowledge [19]. We make no claim that our work is the first application of tangles to quantum information; see §1e for an extensive discussion of related work.

### (ii) Insight

Throughout, the shaded tangle syntax gives a new way to understand why each procedure works. Since the notation is sound, this can be regarded as giving a degree of insight into the structure of the procedures, which we leverage in particular to produce our generalized and novel algorithms. For example, in our verification of error correcting codes, the errors are literally 'trapped by bubbles' and removed from the diagram, and in our verification of cluster chain surgery procedures the qubits are literally untangled from the chain. In both cases, this gives a powerful intuition for these schemes which we believe to be new. This stands in contrast to traditional verification methods in quantum computer science [13], where a procedure is often given as a series of linear maps encoded algebraically (e.g., as matrices of complex numbers), and verification involves composing the maps and examining the result; from this perspective, high-level structure can be difficult to perceive, and it may be unclear whether a procedure can be generalized.

### (iii) Efficiency

Where our methods apply, we can often give the circuit, specification and verification in a concise way; compare for example our discussion of figure 2 with the traditional verification of a related procedure due to Uchida *et al.* [1], which requires a page of algebra, and is also less general. As a consequence, even in this relatively short paper, we are able to give detailed analyses of 13 distinct procedures. We suggest that our methods would therefore be suitable for reasoning about large-scale quantum programs, such as architectures for quantum computers.

## (d) Criticism

### (i) Completeness

We define our semantics to be *sound* if topological isotopy implies computational equivalence, and *complete* if computational equivalence implies topological isotopy. The main semantics we give is sound, allowing the verification method for quantum procedures that we use throughout the paper. However, it is not complete, meaning that there exist quantum procedures that cannot be verified by our methods.[5] Achieving completeness is an important focus of future work. We note that the *ZX calculus* (see §1e), a dominant existing high-level approach to quantum information, shares this property of being sound but not complete [20], although it is complete for the stabilizer fragment [21].

---

[5]Our language is also not *universal*, meaning that not all quantum circuits can be constructed. It would be easy to make it universal by adding additional 1-qubit generators; however, without completeness, this has limited value.

### (ii) Algorithms

All of our examples are in the broad area of quantum communication; we do not study quantum algorithms, such as Grover's or Shor's algorithms [13]. These algorithms have been analysed in the related categorical quantum mechanics (CQM) approach [22]; in future work, we aim to analyse them using our new syntax.

## (e) Related work

### (i) Categorical quantum mechanics

Our work emerges from the CQM research programme, initiated by Abramsky and Coecke [23] and developed by them and others [21,22,24–34], which uses monoidal categories with duals to provide a high-level language for quantum programmes, using in particular a graph-based language called the *ZX calculus* [26,30]. CQM verifications have been given for some procedures related to those that we analyse, including the Steane code [35], and cluster state arguments [26,31]. Many of the advantages of our calculus over traditional techniques—such as the power of the diagrammatic language, and its topological flavour—inherit directly from the CQM programme.

The current authors have previously shown that CQM methods can be extended to a higher-categorical setting [36–38], developing the work of Baez on a categorified notion of Hilbert space [39]. This work develops these ideas, by bringing it into contact with the mathematics of shaded knots.

We give here some important points of distinction between traditional CQM techniques and our present work. Unlike the ZX calculus, our calculus is purely topological, in the sense that equality reduces precisely to topological isotopy; in contrast, the ZX calculus contains a number of algebraic equalities, which do not have a direct topological interpretation. Also, our calculus is incomparable in strength to the ZX calculus, which is restricted (in its basic form) to Clifford quantum theory; neither calculus can simulate the other in general. As a result, we are able to analyse many protocols that have not previously been analysed with ZX methods, as well as discover a number of new and generalized protocols.

### (ii) Statistical mechanics

There is a rich interplay between quantum information (QI), knot theory (KT) and statistical mechanics (SM). The KT–SM and SM–QI relationships are quite well explored in the literature, unlike the KT–QI relationship, which is our focus here.

The KT–SM relationship was first studied by Kauffman, Jones and others [40–42], who showed how to obtain knot invariants from certain statistical mechanical models. Much of the mathematical foundations of our paper are already present in the paper [41], including the shaded knot notation. Work on the SM–QI relationship has focused on finding efficient quantum algorithms for approximating partition functions of statistical mechanical systems [43–46], for which the best-known classical algorithm is often exponential. 'Chaining' these relationships allows one to take a knot, obtain from it a statistical mechanical model, and then write down a quantum circuit approximating the model's partition function, giving overall a mapping from knots to quantum circuits, which ends up closely matching the construction we present. In particular, the relation between Hadamard matrices and shaded tangles, and the specific Hadamard matrices used in this paper are well-known, going back to results of Jones [41] on building link invariants from statistical mechanical models. The novelty here is of course the large range of new and existing quantum procedures which we are able to analyse using this knot-theoretic notation.

The direct KT–QI relationship has also been emphasized by Kauffman and collaborators [47,48], by Jaffe, Liu and Wozniakowski [10,49–52] and also in the field of topological quantum

computing [53], where (as here) a strong analogy is developed between topological and quantum entanglement.

### (iii) Planar algebras

The graphical notation we employ can be described formally as a *shaded planar algebra*, although we do not use that terminology in this paper, preferring a more elementary presentation. The relationship between shaded planar algebras and Hadamard matrices was first suggested by Jones [54], and developed by the present authors [36–38]. In particular, although we are using 2-categorical terminology, all planar diagrams[6] in this work can alternatively be understood in terms of Jones' spin model planar algebra [54, example 2.8].

Recently, Jaffe, Liu and Wozniakowski have described a related tangle-based approach to quantum information based on *planar para algebras* [10,49–52]. In particular, up to a Fourier transform, the Hadamard matrix in figure 8*b* coincides with the one used for the braiding in [52]. Nevertheless, there are several points of distinction that can be drawn between our work and theirs. Firstly, their planar para algebra setting is quite different to the mathematical structure we use. Secondly, their diagrams must be decorated with additional indices encoding measurement results, while our diagrams do not require such indices. Thirdly, there is little overlap between the quantum procedures analysed so far in each setting (although see §5d.).

### (iv) Tangles

The mathematics of tangles have been applied by other authors to computation, including to quantum computation. Carmi and Moskovic develop a theory of tangle machines [55], where tangle diagrams (although not 2-shaded as we use them here) represent networks which can process information; these authors show their ideas apply to adiabatic quantum computation, which is quite different to the circuit model of quantum computation to which our work is most closely connected. Topological quantum computation [56] is a model of quantum computation where computation proceeds by braiding *anyons* in a three-dimensional spacetime; as with our work, this involves linear solutions to a tangle-like calculus, although without the 2-shading or the first Reidemeister move. Kauffman & Buliga [57,58] have studied an intriguing relationship between the lambda calculus and knots, which again has a similar flavour, with knot diagrams representing computations. There has also been some work on using knot-theoretic methods for verification in linear logic ([59] and P-A Melliès 2009, unpublished draft) and separation logic [60]. The precise relation between this work and ours is unclear at present, and an interesting area for future study. Our tangles are equipped with a 2-shading, a structure which has been well studied for links, significantly by Jones in the application to planar algebras [54] mentioned above; see also the excellent survey by Kauffman [61]. We also note the work of Moskovich and Carmi on information fusion [62], where tangle diagrams are used to analyse error correction procedures in a way that may be related to our work in §6 on quantum error correcting codes; understanding these connections is a key area for future work.

## 2. Mathematical foundations

### (a) Graphical calculus

The graphical calculus for describing composition of multilinear maps was proposed by Penrose [63], and is today widely used [24,25,64–66]. In this scheme, wires represent Hilbert spaces and vertices represent multilinear maps between them, with wiring diagrams representing composite linear maps.

In this article, we use a generalized calculus that involves *regions*, as well as wires and vertices; see figure 4*a* for an example. This is an instance of the graphical calculus for symmetric monoidal

---

[6]This excludes the diagrams with overlapping regions in §5 which explicitly use the monoidal structure of the underlying 2-category.

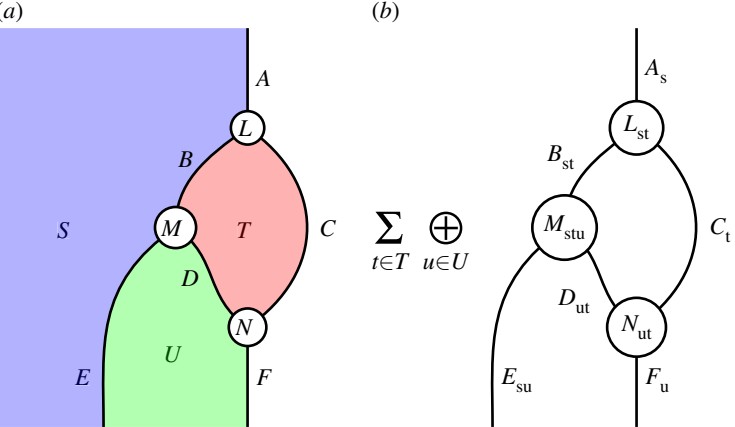

**Figure 4.** The graphical calculus of 2Hilb. (*a*) A diagram in 2Hilb, (*b*) its interpretation for any $s \in S$. (Online version in colour.)

2-categories[7] [67–70] applied to the 2-category **2Hilb** of finite-dimensional 2-Hilbert spaces [39]. The 2-category of 2-Hilbert spaces can be described as follows [37,71]:

— objects are natural numbers;
— 1-morphisms are matrices of finite-dimensional Hilbert spaces;
— 2-morphisms are matrices of linear maps.

We represent composite 2-morphisms in this 2-category using a graphical notation involving regions, wires and vertices, which represent objects, 1-morphisms and 2-morphisms, respectively. In §5, we also use the monoidal structure of **2Hilb**, represented graphically by 'layering' diagrams above each other.

Here we use a particular fragment of this language, which—aside from the monoidal structure—corresponds to the *spin model planar algebra* of Jones [54, example 2.8]. Nonetheless, we emphasize that we use the techniques of monoidal 2-categories here, and not the techniques of planar algebras.

### (i) Elementary description

While these structures are widely used in higher representation theory, they are not yet prevalent in the quantum computing community. To help the reader understand these new concepts, we also give a direct account of the formalism in elementary terms, that can be used without reference to the higher categorical technology (see also [36]).

In this direct perspective, regions are labelled by *finite sets*. Wires and vertices now represent *families* of Hilbert spaces and linear maps, respectively, indexed by the elements of the sets labelling all adjoining regions. A composite surface diagram represents a family of composite linear maps, indexed by the elements of all regions open on the left or right. For regions open only at the top or bottom of the diagram, we take the direct sum over elements of the indexing set, while for closed regions, we take the vector space sum over elements of the indexing set.

We give an example in figure 4. In the diagram on the left, regions are labelled by finite sets $(S, T, U)$, with unshaded regions labelled implicitly by the 1-element set; wires are labelled by families of finite-dimensional Hilbert spaces $(A, B, C, D, E, F)$; and vertices are labelled by families of linear maps $(L, M, N)$. For wires and vertices, the families are indexed by the sets associated with all neighbouring regions: for example, for $s \in S$ and $t \in T$, we have Hilbert spaces $A_s$, $B_{st}$ and $C_t$, and $L_{st} : B_{st} \otimes C_t \to A_s$ is a linear map. The single diagram on the left represents an entire

---

[7]Here and throughout, we use the term '2-category' to refer to the weak structure, which is sometimes called 'bicategory'.

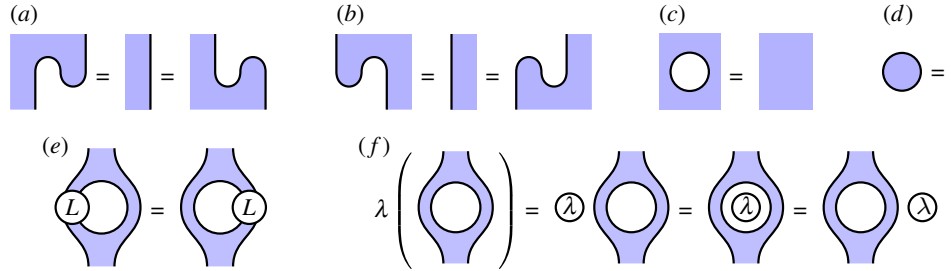

**Figure 5.** Some identities in the graphical calculus. (Online version in colour.)

family of linear maps, with the maps comprising this family given by the right-hand diagram for different values of $s \in S$. We take the direct sum over index $u \in U$, since its region is open only at the bottom of the diagram, and the vector space sum over index $t \in T$, since its region is closed. This description also applies to situations where one region is layered above another; formally, such diagrams are constructed using the monoidal structure of **2Hilb**, and they do not fundamentally complicate the nature of the calculus.

Given this interpretation of diagrams $D$ as families of linear maps $D_i$, we define two diagrams $D, D'$ to be *equivalent* when all the corresponding linear maps $D_i, D'_i$ are equal; we define the *scalar product* $\lambda D$ as the family of linear maps $\lambda D_i$; we define the *adjoint* $D^\dagger$ as the family of adjoint linear maps $(D_i)^\dagger$; and we say that $D$ is unitary if all the maps $D_i$ are unitary. Following convention [66], we depict the adjoint of a vertex by flipping it about a horizontal axis.

Throughout this paper, we only use a highly restricted part of **2Hilb**; it is this restricted part that agrees with the spin model planar algebra, as we mention above. Every shaded region we assume to be labelled by a single fixed finite set $S$. All wires bound precisely one shaded region and one unshaded region, and these wires are always labelled by a family of one-dimensional Hilbert spaces $\mathbb{C}$. Nonetheless, the calculus is not trivial. For example, we can build the identity on a non-trivial Hilbert space as the diagram figure 1$a$ under the rules set out above, this is the identity map on $\oplus_{s \in S}(\mathbb{C} \otimes \mathbb{C}) \simeq \mathbb{C}^{|S|}$.

Also, we add the following components to our language. In the first case, there is an op en region, and we use the obvious isomorphism $\mathbb{C} \simeq \mathbb{C} \otimes \mathbb{C}$ to build the associated families of linear maps.

$$\forall s \in S, \ \mathbb{C} \simeq \mathbb{C} \otimes \mathbb{C} \qquad\qquad 1 \mapsto \sum_{i \in S} |i\rangle \tag{2.1}$$

Flipping these components about a horizontal axis denotes the adjoint of these maps, as discussed above. With these definitions, the equations illustrated in figure 5 can be demonstrated; in that figure, the vertex $L$ and the scalar $\lambda \in \mathbb{C}$ are arbitrary.

## (b) Shaded tangles

The *Reidemeister moves* [61, ch. 1] are the basic relations of classical knot theory. In this section, we present an equational theory of shaded knots, which use shaded versions of the Reidemeister moves. This theory is related to work of Jones [41] on invariants of shaded links from statistical mechanical models, and to work of other authors as discussed in §1e.

We begin by supposing the existence of the following *shaded crossing* vertices, depicted as follows:

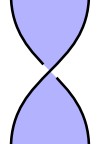 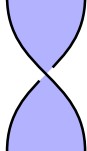 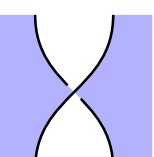 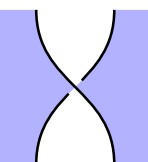

$$\tag{2.2}$$

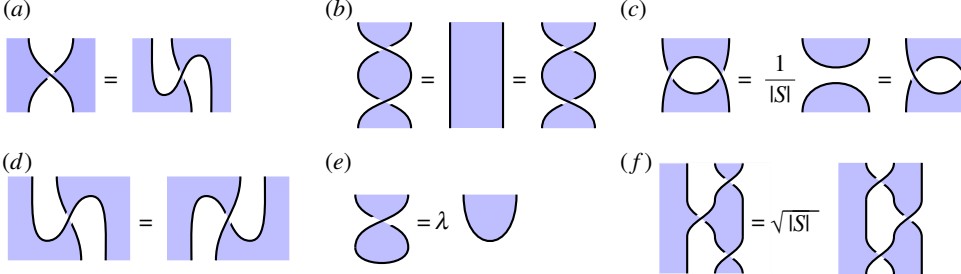

**Figure 6.** The shaded tangle calculus. (Online version in colour.)

The basic identities of figure 5 are assumed to hold. We say that these crossings satisfy the *basic calculus* when it satisfies the equations of figure 6*a*–*d*, and the *extended calculus* when it additionally satisfies equation figure 6*e*–*f*.[8] A *shaded tangle diagram* is a diagram constructed from the components of this calculus, the shaded cups (2.1), and their adjoints.

This is the formal definition of our calculus, as a subset of the graphical calculus of the monoidal 2-category **2Hilb**. Note the background features of the calculus as described in §2a still apply, including the basic identities of figure 5, and the rule that the adjoint of a vertex is depicted by flipping that vertex about a horizontal axis. In particular, this implies that of the four crossings displayed as (2.2) above, the first two are adjoint, and the last two are adjoint. For our more complex applications later in the paper, we may also use more advanced features of the graphical calculus of **2Hilb**; in all cases, including diagrams with overlapping regions, this can be understood formally in terms of the elementary description given in §2a.

The extended calculus has the following attractive property. This is straightforward, and we do not claim it is original; for related work, see Kauffman [61] and Cordova *et al.* [72].

**Theorem 2.1.** *Two shaded tangle diagrams with the same upper and lower boundaries are equal under the axioms of the extended calculus (up to an overall scalar factor) just when their underlying tangles, obtained by ignoring the shading, are isotopic as classical tangles.*

We emphasize that there is no corresponding full isotopy statement for the basic calculus, since it only satisfies a subset of the shaded Reidemeister moves.

In **2Hilb**, we can classify representations of the basic calculus as follows. Note that from the discussion of §2a, the first vertex indicated in (2.2) represents in **2Hilb** a linear map of type $\mathbb{C}^{|S|} \to \mathbb{C}^{|S|}$, and is therefore canonically represented by a matrix, which we assume to have matrix entries $H_{a,b}$.

Recall that a *Hadamard matrix* is a unitary matrix with all coefficients having the same absolute value. A Hadamard matrix $\{H_{a,b}\}_{a,b}$ is *self-transpose* if $H_{a,b} = H_{b,a}$ for all $a, b$.

**Theorem 2.2.** *In **2Hilb**, a shaded crossing yields a solution of the basic calculus just when it is equal to a self-transpose Hadamard matrix.*

The following theorem identifies the additional constraint given by the extended calculus.

**Theorem 2.3.** *In **2Hilb**, a self-transpose Hadamard matrix satisfies the extended calculus just when:*

$$\sum_{r=0}^{|S|-1} \bar{H}_{ar} H_{br} H_{cr} = \sqrt{|S|}\, \bar{H}_{ab} \bar{H}_{ac} H_{bc}. \tag{2.3}$$

Proofs of theorems 2.1–2.3 are given in appendix B.

A full classification of representations of this extended calculus is not known. However, it is known that solutions exist in all finite dimensions; we present this in appendix A.

---

[8]In presenting this calculus, $\lambda$ is an arbitrary non-zero constant, and we implicitly use the rule described in §2a regarding the representation of the adjoint as a reflected diagram, which causes the crossing type to change. This calculus also defines a rotated crossing in figure 6*a*.

**Figure 7.** Explicit expressions for the 1- and 2-qudit gates. (*a*) 1-qudit gate, (*b*) adjoint 1-qudit gate, (*c*) 2-qudit gate, (*d*) adjoint 2-qudit gate. (Online version in colour.)

## (c) Circuits and specifications

### (i) Scalar factors

From this point onwards, we drop the scalar factors appearing in the shaded tangle calculus, since they complicate the diagrams. More formally, every component we use in the remainder of the paper is proportional to an isometry, and we silently replace it with its isometric equivalent.

### (ii) Circuits

We write our quantum circuits in terms of four basic components of this shaded tangle language.

— *Qudits.* As mentioned above, figure 1*a* is interpreted as the identity map on $\mathbb{C}^{|S|}$, some finite-dimensional Hilbert space. This gives us our qudit.

— *Qudit preparations.* In expression (2.1), we draw a blue 'cup' to indicate the state $\sum_{i \in S} |i\rangle \in \mathbb{C}^{|S|}$, which we interpret as a *qudit preparation* (figure 1*b*.)

— *Qudit gates.* Given that the first vertex of (2.2) corresponds to a self-transpose Hadamard with matrix entries $H_{ij} = H_{ji}$, we obtain concrete representations for our 1- and 2-qubit gates, given in figure 7.

### (iii) Specifications

We can write our specifications using arbitrary shaded tangle diagrams—that is, using the cups from (2.1) and their adjoint caps, as well as arbitrary shaded crossings. Caps need to be excluded when describing circuits since they are not (proportional to) isometries and are therefore not directly interpretable as circuit components. This does not prevent us using them in specifications, however, since these will not be directly executed; they exist only to define the mathematical behaviour of the overall procedure.

### (iv) Examples

We give some concrete examples of our basic circuit components. Some procedures we analyse require only the basic calculus to be satisfied, in which case we can choose any self-transpose Hadamard matrix as our basic data; it is interesting to flag when this is the case, since these have a much broader class of models. A standard choice is the *qubit Fourier Hadamard*, which satisfies the basic calculus but not the extended calculus, illustrated in figure 8*a*. Other

**Figure 8.** Different examples of our circuit elements. (Online version in colour.)

procedures require a Hadamard representing the extended calculus; an example is the metaplectic Hadamard illustrated in figure 8*b* constructed using the methods of appendix A. This Hadamard has been used in the cluster state literature for neighbourhood inversion on a cluster graph [4, proposition 5], an operation we verify in §4e for a linear graph.

# 3. Entangled states

In this section, we describe several forms of entanglement and their representations in our graphical calculus.

## (a) GHZ states

*GHZ states* were introduced by Greenberger *et al.* [73] to give a simplified proof of Bell's theorem. We define the unnormalized *n*-partite qudit GHZ state as follows:

$$|\text{GHZ}_n\rangle := \sum_{k=0}^{d-1} |k, \ldots, k\rangle. \tag{3.1}$$

**Proposition 3.1.** *GHZ states are represented as follows:*

$$|\text{GHZ}_n\rangle = \tag{3.2}$$

This proposition follows from appropriately composing the expressions for the cups (2.1) according to the graphical calculus of **2Hilb** outlined in §2a. Alternatively, this also arises directly from the representation of GHZ states in the CQM programme [28]. Important special cases for qubits are the $|+\rangle$ state $|\text{GHZ}_1\rangle = |0\rangle + |1\rangle$, and the Bell state $|\text{GHZ}_2\rangle = |00\rangle + |11\rangle$.

## (b) Cluster chains

Another important class of entangled states are the *cluster states* or *graph states* [4,5,74] and their qudit generalizations associated with Hadamard matrices [2]. Cluster states have numerous applications, most prominently in the theory of measurement-based quantum computation [3,75] and quantum error correction [74]. Here we will focus on qudit *cluster chains*, cluster states entangled along a chain.

Given a self-transpose *d*-dimensional Hadamard matrix *H*, the *n*-partite qudit cluster chain associated with *H* is the following, where we conjugate the matrix due to our conventions:

$$|C_n\rangle := \sum_{a_1,\ldots,a_n=0}^{d-1} \bar{H}_{a_1,a_2} \cdots \bar{H}_{a_{n-1},a_n} |a_1 \cdots a_n\rangle. \tag{3.3}$$

**Figure 9.** A tangle gate and a tangle state. (Online version in colour.)

**Proposition 3.2.** *Cluster chains are represented as follows*:

$$|C_n\rangle = \qquad\qquad\qquad\qquad\qquad\qquad\qquad\qquad\qquad (3.4)$$

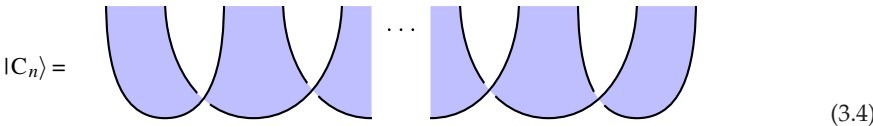

This is essentially the same as the representation of cluster states used in the CQM programme [26,31]. If all Hadamard matrices in (3.3) are the Fourier Hadamard, then this recovers conventional cluster chains [5].

### (c) Tangle gates and tangle states

More generally, a *tangle gate* is any circuit built from 1- and 2-qudit gates and their adjoints, and a *tangle state* is a tangle with no inputs built from qudit preparations and tangle gates (figure 9). Such tangle states and gates can be arbitrarily complex, and have all the algebraic richness of knot topology. If a Hadamard represents the extended calculus, then two tangle states or gates are equal just when the corresponding tangles are isotopic, as established by theorem 2.1.

## 4. Manipulating quantum states

In this section, we verify a wide variety of procedures for creating and manipulating entangled states, including a new program for robust state transfer within a cluster chain-based quantum computer.

### (a) Constructing GHZ states (figure 2)

*Overview.* We can use our formalism to design and verify a procedure for constructing $n$-partite GHZ states.

*Circuit figure 2a.* Begin by preparing $n$ qudits, then apply a sequence of 2- and 1-qudit gates as indicated in figure 2*a* for $n = 3$ qudits.

*Specification figure 2b.* This is the 3-qudit instance of (3.2), the tangle state corresponding to a GHZ-state.

*Verification.* Immediate by isotopy: the middle and rightmost qudit preparations in figure 2*a* move up and left, underneath the diagonal strand, producing figure 2*b*.

*Calculus.* This requires only the basic calculus.

*Novelty.* The GHZ version is known for the qubit Fourier Hadamard and was described very recently [1] for the *qudit Fourier matrices* $H_{ab} = e^{(2\pi i/d)ab}$. For the self-transpose qudit Hadamard case covered here, the procedure seems new.

### (b) Creating cluster chains

*Overview.* Analogously to §4a, we can consider the design of a circuit to create a cluster chain.

*Circuit.* We illustrate this in (3.4): we begin with $n$ qudit preparations, then perform 2-qudit gates a total of $n - 1$ times.

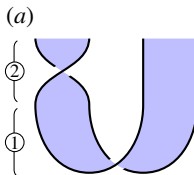 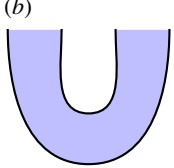

*(a)*  *(b)*

**Figure 10.** Converting a 2-party cluster state into a GHZ state. (*a*) Circuit and (*b*) specification. (Online version in colour.)

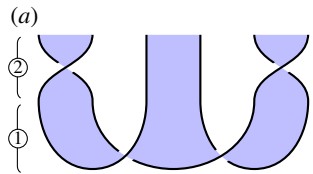 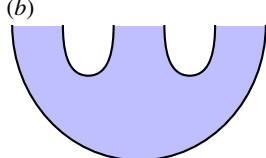

*(a)*  *(b)*

**Figure 11.** Converting a 3-party cluster state into a GHZ state. (*a*) Circuit and (*b*) specification. (Online version in colour.)

*Specification*. We also take expression (3.4) to be the specification, proposition 3.2 shows that this recovers the standard algebraic definition (3.3).

*Verification*. Trivial, the circuit and specification are equal.

*Novelty*. This procedure is well known for conventional and generalized cluster states [2,3]. Our treatment here is not fundamentally different to the CQM analysis of Coecke, Duncan and Perdrix [26,31].

## (c) Local unitary equivalence (figures 10 and 11)

*Overview*. In the case of 2 or 3 parties, cluster chains can be converted into GHZ states by applying 1-qudit gates on certain sites. This means that, in a strong sense, they are equivalent computational resources. The reverse process, converting GHZ states to cluster chains, could be just as easily described.

*Circuit*. For 2 and 3 parties, we illustrate the circuits in figures 10*a* and 11*a*, respectively. ① Construct a cluster state. ② Perform a 1-qubit gate at certain sites.

*Specification*. Illustrated in figures 10*b* and 11*b*, these are instances of the general GHZ specification (3.2).

*Verification*. Immediate by isotopy. For the 2-party case, a loop of string in the lower-left of figure 10*a* contracts to the top of the diagram, giving figure 10*b*. For the 3-party case, we perform similar contractions for loops of string at the lower-left and lower-right of figure 11*a*, which move above and below a third strand, respectively, giving figure 11*b*.

*Calculus*. This requires only the basic calculus.

*Novelty*. This is known both for conventional [5] and generalized [2] cluster chains. For more than three parties, it is known to be false, and indeed our method fails in these instances.

## (d) Cutting cluster chains (figure 12)

*Overview*. Given a cluster chain of length $n$ we can *cut* a target node from the chain, yielding two chains of total length $n - 1$ and the target node in the $|+\rangle$ state.[9]

*Circuit figure 12a*. ① Prepare a cluster state, of which only a central part is shown. ② Perform two adjoint 2-qudit gates both involving a central target qudit.

---

[9]In some variants the target node is instead destroyed by a projective measurement, and controlled operations performed on the adjacent qudits [4, §3]; the mathematical structure is identical to the version we analyse. A similar comment applies to the splicing procedure of §4e.

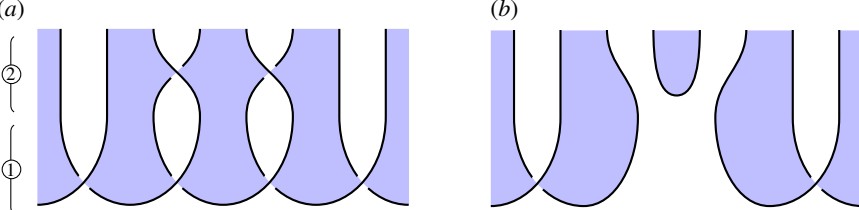

**Figure 12.** Cutting a cluster chain. (*a*) Circuit and (*b*) specification. (Online version in colour.)

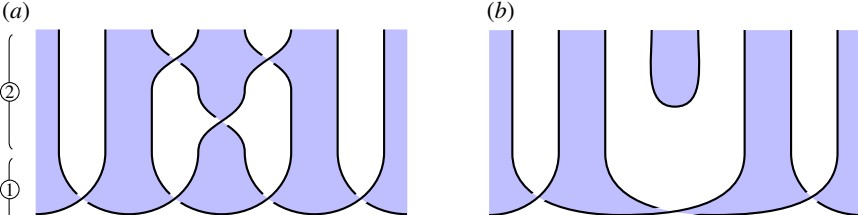

**Figure 13.** Splicing a cluster chain. (*a*) Circuit and (*b*) specification. (Online version in colour.)

*Specification figure 12b*. Prepare two separate cluster chains, and separately prepare the target node in the $|+\rangle$ state.

*Verification*. Immediate by isotopy: starting with figure 12*a*, we cancel the inverse pairs of crossings on the left and right of the central qudit, yielding figure 12*b*.

*Calculus*. This requires only the basic calculus.

*Novelty*. For the qubit Fourier Hadamard matrix, this is well known; see [6] and [4, §3]. Here, and in the next subsection, our verification provides new insight into how these chain manipulation programmes work.

## (e) Splicing cluster chains (figure 13)

*Overview*. Given a cluster chain of length $n$, we can *splice* a target node from the chain, yielding a single chain of length $n - 1$, and the target node in the $|+\rangle$ state.

*Circuit figure 13a*. ① Prepare a cluster state, of which only a central part is shown. ② Perform a 1-qudit gate on the target qudit, and then two 2-qudit gates involving the target qudit and each of its adjacent qudits.

*Specification figure 13b*. Prepare a cluster chain of length $n-1$, and separately prepare the target node in the $|+\rangle$ state.

*Verification*. By isotopy, although harder to see by eye than previous examples. Looking closely, one can see that the target qudit in figure 13*a* is unlinked from the other strings, and so the entire shaded tangle can be deformed to give figure 13*b*.

*Calculus*. This requires the extended calculus.

*Novelty*. It is well known that certain local operations on cluster chains splice the chain (*neighbourhood inversion* on graph states, see [6] and [4, Prop. 5]); our analysis is more general since it applies for any qudit Hadamard satisfying the extended calculus. The standard procedures use cluster chains based on the qubit Fourier Hadamard, and require additional phase corrections, which effectively serve to convert the Hadamard into one representing the extended calculus. We avoid this by building the cluster chain itself from a Hadamard representing the extended calculus.

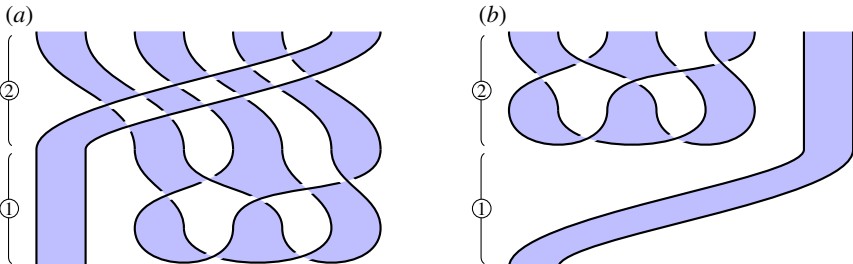

**Figure 14.** Cluster-based quantum state transfer. (*a*) Circuit and (*b*) specification. (Online version in colour.)

## (f) Cluster-based quantum state transfer (figure 14)

*Overview.* In real quantum computing architectures that make heavy use of cluster states, such as the ion trap model [76], qubits are encoded in individual atomic structures, often arranged in a linear chain. One may want to move a target qubit to a different position in the chain—for example, to enable a multi-qubit gate to be applied, or to put the target qubit into position to be measured—but physically moving individual atoms may be impractical [77], and the 2-qubit swap gate $|ij\rangle \mapsto |ji\rangle$ may be hard to implement.

Here we introduce a state transfer program for moving a target qubit along a cluster chain, which uses only the tangle interaction used for generating the cluster states which the machine may be optimized to perform, and which is robust against tangle gate errors on the non-target qubits.

*Circuit figure 14a.* ① Begin with a target qudit on the left, and a cluster state to the right, corrupted with an arbitrary tangle gate. ② Perform a repeating sequence of 1- and 2-qudit tangle gates along the chain as indicated.

*Specification figure 14b.* ① Move the target qudit to the rightmost position. ② Recreate the cluster state with tangle gate error on the remaining qudits.

*Verification.* Immediate by isotopy: the tangle state in the lower-right of figure 14*a* moves up and left, underneath the diagonal wires, producing the shaded tangle figure 14*b*.

*Calculus.* This requires the extended calculus.

*Novelty.* We believe this procedure is new.

## 5. Teleportation

Teleportation is a major theme in quantum information, playing an important structural role in the design of quantum computers. Here we use our topological calculus to verify a wide range of teleportation protocols. Our analysis in this section requires some small modifications to our graphical language, which we briefly describe.

*Partitions.* In this section, it will often be important that the resources are *partitioned*, with qudits controlled by a number of different agents. Where appropriate we use vertical dashed lines to indicate this partitioning as an informal visual aid.

*Measurement.* It is standard that if a qudit is used only as the control side of a controlled 2-qudit unitary, then it may be considered as having been *measured*, and the controlled unitary interpreted as a classically controlled family of 1-qubit gates [13, Exercise 4.35]. To indicate this in our formalism, we shade the corresponding qudit red, and interpret it as a classical dit. As with partitions above, this is an informal visual aid; with respect to the mathematics, the red and blue shaded regions are equivalent. When a red region meets a vertical dashed partitioning line, we interpret this as classical communication.

*Overlapping.* When multiple red regions exist at once, we sometimes draw them as *overlapping*. This will necessitate the use of new sorts of crossings, as shown in figure 15*a*. Unlike the crossings

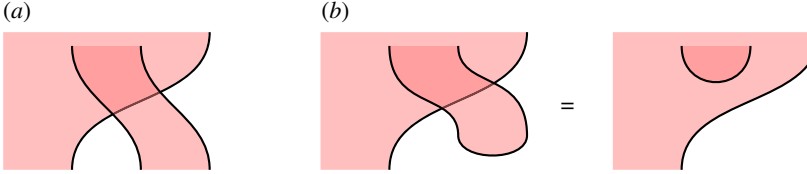

**Figure 15.** Graphical representation of the monoidal structure on 2Hilb. (*a*) Overlapping regions. (*b*) An isotopy using the monoidal structure. (Online version in colour.)

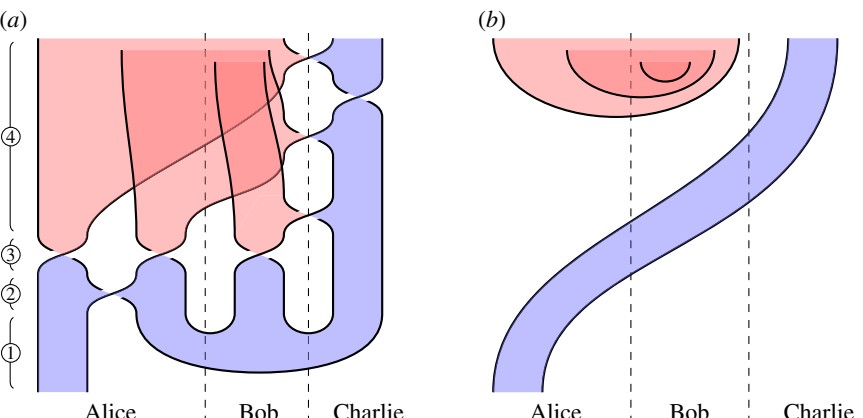

**Figure 16.** Measurement-based GHZ teleportation. (*a*) Circuit and (*b*) specification. (Online version in colour.)

of blue regions which represent Hadamards, these crossings of red regions are mathematically trivial, and simply encode the reordering of classical data. In terms of our categorical semantics, this overlapping is described by the monoidal structure of the 2-category, and this is the foundation for our approach; it is related to the *virtual knots* of Kauffman and others [78], which also have two kinds of crossing. The local moves of the monoidal 2-category structure are illustrated in figure 15*b*; they allow structures in different overlapping sheets to move freely past each other. Here we treat aspects of virtual shaded knot theory, and the corresponding Reidemeister moves, informally; our mathematical model nonetheless remains precise, as it continues to be an instance of the graphical calculus of **2Hilb** as a monoidal 2-category, as described in figure 4.

## (a) Measurement-based GHZ teleportation (figure 16)

*Overview.* Teleport a state from agent 1 to agent $n$ using a shared $n$-partite GHZ state and classical communication, with all corrections performed by agent $n$. (Note relationship to §5b.)

*Circuit figure 16a.* We illustrate the procedure for three agents: Alice, Bob and Charlie. ① Alice has a qudit to be teleported, and all agents share a GHZ state, perhaps generated according to §§4a. ② Alice applies a 2-qudit gate. ③ Alice and Bob measure all their qudits in the complementary basis determined by the Hadamard, and send the results classically to Charlie. ④ Charlie performs unitaries on his qudit, dependent on Alice and Bob's measurement results.

*Specification figure 16b.* Alice's qudit is passed to Charlie, and the classical dits are produced by measuring $|+\rangle$ states.

*Verification.* By isotopy, the three 'cups' forming the GHZ state can be pulled up one at a time.

*Calculus.* This requires only the basic calculus.

*Novelty.* This protocol was first described by Karlssen [9] for the qubit 3-party case, by Hillery [8] for the qubit $n$-party case and by Grudka [7] for qudit Fourier matrices. For general self-transpose Hadamards this seems new.

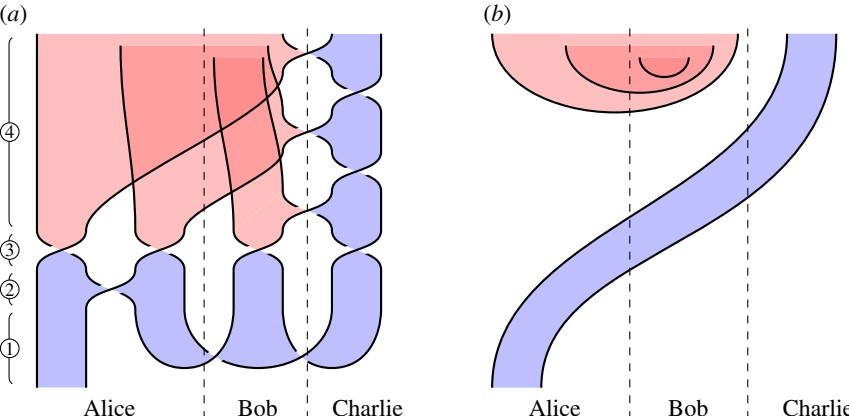

**Figure 17.** Measurement-based cluster chain teleportation. (*a*) Circuit and (*b*) specification. (Online version in colour.)

*Discussion.* This procedure can be understood as distributing Alice's initial state across all parties. Only after all parties cooperate and reveal their measurement results can Alice's state be reconstructed by Charlie. From this perspective, the procedure is reminiscent of a secret sharing protocol, and it has been discussed in these terms by Hillery [8].

## (b) Measurement-based cluster chain teleportation (figure 17)

*Overview.* Teleport a state from agent 1 to agent $n$ using a shared $n$-party cluster chain and classical communication, with all corrections performed by agent $n$. (Note relationship to §5a.)

*Circuit figure 17a.* Almost identical to the circuit of §5a, except with an initial cluster chain rather than GHZ state, and Charlie's steps are slightly modified.

*Specification figure 17b.* Alice's qudit is passed to Charlie, and the classical dits are produced by measuring $|+\rangle$ states.

*Verification.* Similar to §5a.

*Calculus.* This requires only the basic calculus.

*Novelty.* This procedure is known in the case of conventional qubit cluster states [3,4]. We believe it is new for the generalized cluster chains considered here, based on arbitrary self-transpose Hadamards, and in fact a further generalization to arbitrary Hadamards is straightforward.

## (c) Robust GHZ teleportation (figure 18)

*Overview.* Given a chain of $n$ agents sharing a GHZ resource state, teleport a qudit from agent 1 to $n$, in a way which is robust against a large class of errors in the resource state.

*Circuit figure 18a.* We illustrate the procedure for three agents Alice, Bob and Charlie. ① Alice begins with a qudit to be teleported, and all three agents share a GHZ state, perhaps generated according to §4a. ② An error in the form of an arbitrary tangle gate (see §3c) acts on part of the GHZ state as shown. ③ Alice performs a 2-qudit gate, then measures her qudits in the complementary basis determined by the Hadamard, sending the left and right qudit results to Bob and Charlie respectively. ④ Bob applies a controlled unitary to his qudit, then measures his qudit in the complementary basis, and sends his result to Charlie. ⑤ Charlie performs unitaries on his qudit dependent on Alice's and Bob's results.

*Specification figure 18b.* Alice's qudit is passed to Charlie, and the classical dits are produced by applying a tangle gate (in fact, the shading-reversal of the tangle gate used in figure 18a) to $|+\rangle$ states and measuring in the computational basis.

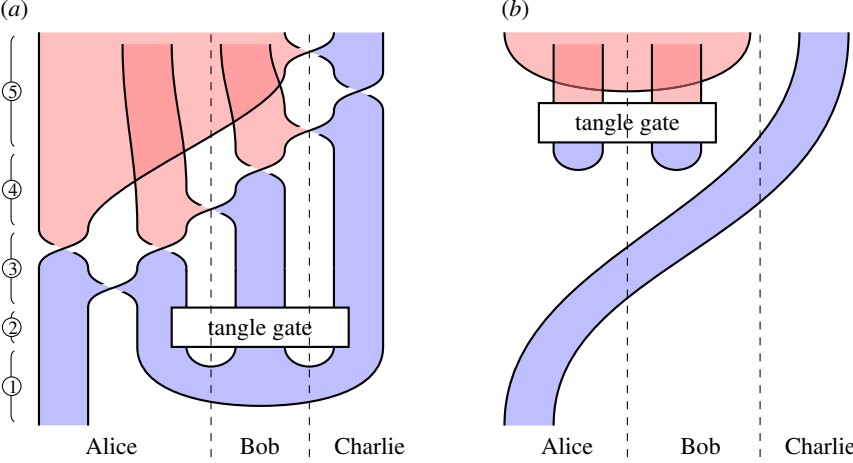

**Figure 18.** Robust GHZ teleportation. (*a*) Circuit and (*b*) specification. (Online version in colour.)

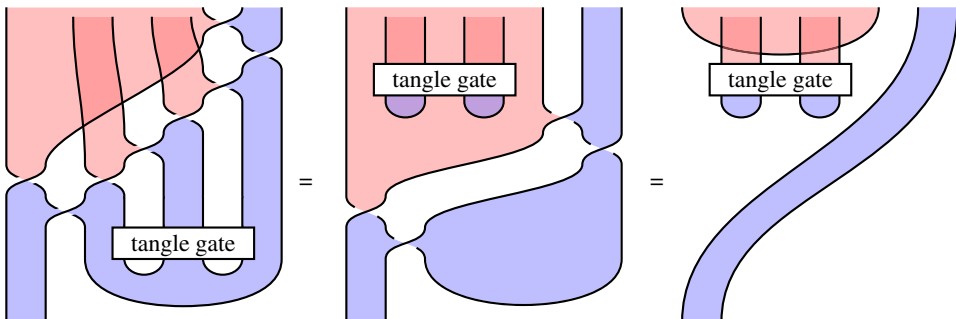

**Figure 19.** The verification isotopy of the robust GHZ teleportation procedure. (Online version in colour.)

*Verification figure 19*. By isotopy, the entire tangle error can be pulled up, 'underneath' the lower diagonal strand, inverting its shading. The lower 'cup' of the GHZ state can then be pulled up similarly.

*Calculus*. The GHZ teleportation procedure requires the basic calculus, and robustness under tangle errors additionally requires equation figure 6*f* of the extended calculus (to allow arbitrary tangle errors, which might themselves involve crossings, to move upwards).

*Novelty*. Ignoring the robustness property, we believe this procedure to be folklore; note that for two parties it corresponds to ordinary Bell state teleportation, and the measure-correct pattern repeated here serves to convert $|GHZ_n\rangle$ to $|GHZ_{n-1}\rangle$. The generalization here to arbitrary self-transpose qudit Hadamards, and (with the extended calculus) the robustness property, seems to be new.

## (d) Non-local controlled unitaries (figure 20)

*Overview*. Suppose that Alice and Bob have separate qudits, and they want to perform a 2-qudit controlled unitary of the following form, where Bob's qudit is the control:

$$C = \sum_{i=0}^{d-1} U_i \otimes |i\rangle \langle i|.$$

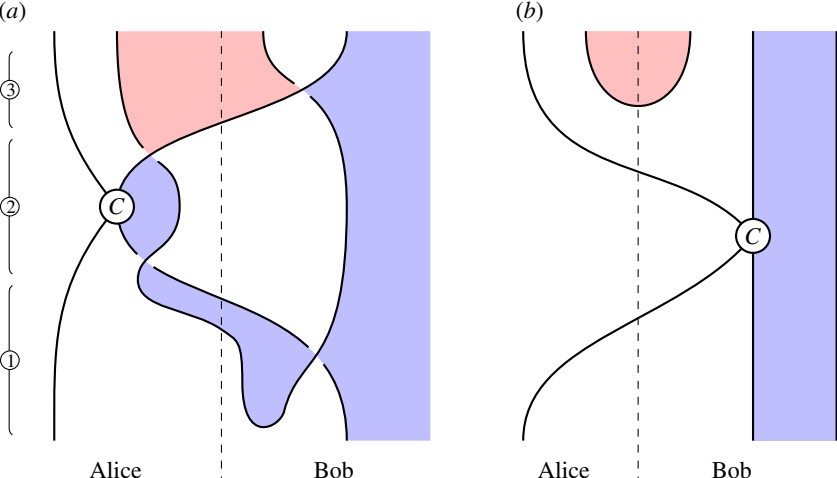

**Figure 20.** Execution of a non-local controlled unitary. (*a*) Circuit and (*b*) specification. (Online version in colour.)

Both qudits are to be kept coherent throughout. The naive solution would be for one party to transport their system to the other party; for the 2-qudit unitary to be performed; and for the system to then be transported back. We describe a protocol to achieve this task with only one quantum transport required.

*Circuit figure 20a*. ① Bob prepares a $|+\rangle$ state, entangles it with his qudit and transports it to Alice. ② Alice performs a 1-qubit gate, performs the controlled unitary $C$, measures in the complementary basis, then sends the result to Bob. ③ Bob performs a correction.

*Specification figure 20b*. Alice's qudit is transported to Bob, who performs the controlled unitary, then passes it back. The classical dit arises from measuring a $|+\rangle$ state.

*Verification*. Immediate by isotopy.

*Calculus*. This requires only the basic calculus.

*Novelty*. A version of this procedure based on qudit Fourier Hadamards and involving an initial teleportation step was considered by Yu *et al.* [11] and described graphically by Jaffe *et al.* [10]. We generalize this here to arbitrary self-transpose Hadamard matrices. However, Jaffe *et al.* describe a different generalization to multiple agents, which we cannot capture.

# 6. Quantum error correction

We now apply our shaded tangle calculus to the theory of quantum error correction. We give a graphical verification of the phase and Shor codes, and we give a substantial new generalization of both based on UEBs. This verification is based on the Knill–Laflamme theorem [79], a powerful theorem in quantum information which establishes a correspondence between error correcting codes and mathematical properties of the encoding isometry. Given an encoding map satisfying these properties, a full error correcting protocol can be constructed.

## (a) Basic definitions

We begin by establishing notation. For $n, k, p, d \in \mathbb{N}$, an $[[n, k, p]]_d^\varepsilon$ code uses $n$ physical qudits to encode $k$ logical qudits, in a way which is robust against errors occurring on at most $\lfloor (p-1)/2 \rfloor$ physical qudits, such that each error is drawn from the subgroup $\varepsilon \subseteq U(d)$. We will be concerned with two types of errors: *full qudit errors*, for which $\varepsilon = U(d)$, and *phase errors*, for which $\varepsilon = P \subset U(d)$, the subgroup of diagonal unitary matrices. The Knill–Laflamme theorem [79] gives a way to identify these codes.

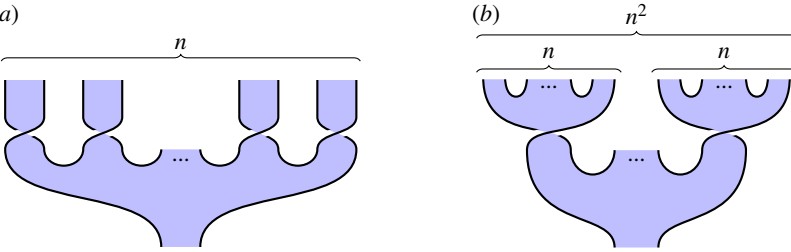

**Figure 21.** The encoding maps $i$ for the phase and Shor codes. (*a*) Phase code and (*b*) Shor code. (Online version in colour.)

**Definition 6.1.** An operator $e : (\mathbb{C}^d)^n \to (\mathbb{C}^d)^n$ is $(p, \varepsilon)$-*local* when it is of the form $e = U_1 \otimes \cdots \otimes U_n$, such that $U_i \in \varepsilon$ for all $1 \le i \le n$, and such that at most $p - 1$ of the operators $U_i$ are not the identity.

**Theorem 6.2 (Knill–Laflamme [79]).** *An isometry* $i : (\mathbb{C}^d)^k \to (\mathbb{C}^d)^n$ *gives an* $[[n, k, p]]_d^\varepsilon$ *code just when, for any* $(p, \varepsilon)$-*local operator* $e : (\mathbb{C}^d)^n \to (\mathbb{C}^d)^n$, *the composite*

$$(\mathbb{C}^d)^k \xrightarrow{i} (\mathbb{C}^d)^n \xrightarrow{e} (\mathbb{C}^d)^n \xrightarrow{i^\dagger} (\mathbb{C}^d)^k \tag{6.1}$$

*is proportional to the identity.*

Informally, the Knill–Laflamme theorem says that we have a $[[n, k, p]]_d^\varepsilon$ code just when, if we perform the encoding map, then perform a $(p, \varepsilon)$-local error, then perform the adjoint of the encoding map, the result is proportional to our initial state. To be clear, any proportionality factor is allowed, even 0.

## (b) Representing errors

In the graphical notation of **2Hilb** (see §2a), in the completely general case, arbitrary qudit phases (that is, diagonal linear maps $\mathbb{C}^n \to \mathbb{C}^n$) and qudit gates (that is, linear maps $\mathbb{C}^n \to \mathbb{C}^n$) are represented as vertices of the following types, respectively:



We draw them in red, as they are interpreted here as errors.

## (c) The phase code

*Overview.* We present a $[[n, 1, n]]_d^{\mathcal{P}}$ code: that is, a code which uses $n$ physical qudits to encode 1 logical qudit in a way that corrects $\lfloor (n-1)/2 \rfloor$ phase errors on the physical qudits. The data are a family of $n$ $d$-dimensional Hadamard matrices.

*Circuit.* The encoding map $i$ is depicted in figure 21*a*.

*Specification.* Satisfaction of the conditions of theorem 6.2.

*Verification.* In figure 22, we illustrate the $n = 3$ version of the code. We must therefore show that the composite $i^\dagger \circ e \circ i$, for any 3-local phase error in which 2 qudits are corrupted by arbitrary phases, is proportional to the identity. Given the symmetry of the encoding map, there are two cases: the errors can occur on adjacent or non-adjacent qudits. We analyse the case of adjacent errors here; the verification for non-adjacent errors is analogous. In the first image of figure 22, we represent the composite $i^\dagger \circ e \circ i$, using some artistic licence to draw the closed curves as circles.

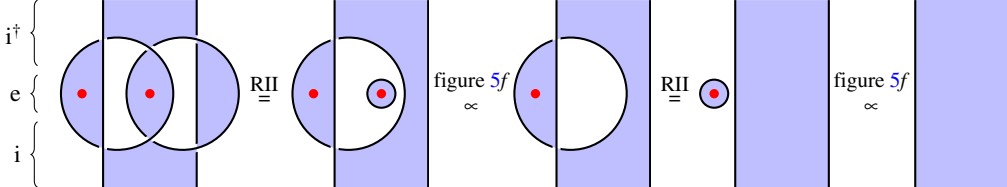

**Figure 22.** Verification of the $[[3, 1, 3]]_d^{\mathcal{P}}$ phase code. (Online version in colour.)

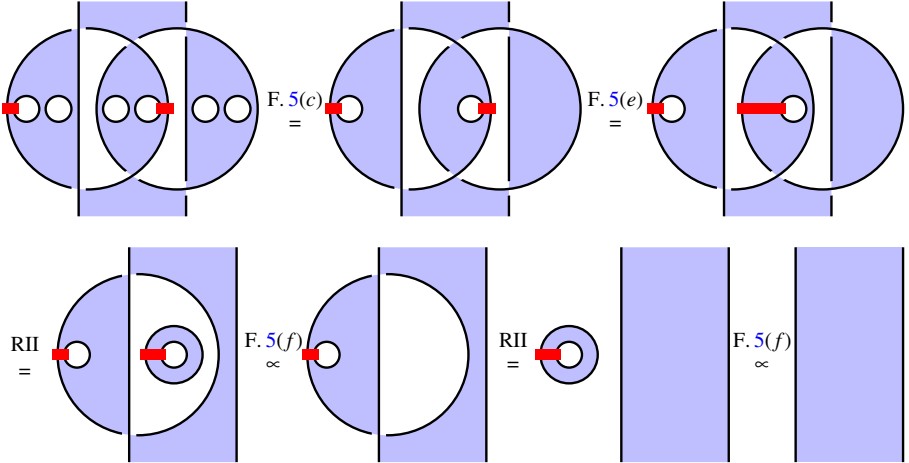

**Figure 23.** Verification of the $[[9, 1, 3]]_d^{U(d)}$ Shor code. (Online version in colour.)

We apply RII moves to cause the errors to become 'captured' by bubbles floating in unshaded regions, which therefore (figure 5f) give rise to overall scalar factors.

*Calculus*. This requires only the basic calculus.

*Novelty*. A major novel feature is the visceral sense of how the protocol works that figure 22 conveys: the phase errors are 'captured by bubbles' and turned into scalar factors. We believe this intuition has not been described elsewhere. At present, we are not able to say what deeper significance this might have. However, it seems robust enough to be applied in other settings, for example, in higher dimension.

In terms of the mathematics, for the qubit Fourier Hadamard, this code is well known [13,14]. The generalization to arbitrary qudit Hadamard follows from work of Ke [12]. Our treatment reveals a further generalization: each of the $n$ Hadamards used to build the encoding map $i$ may be *distinct*, since throughout the verification, we only ever apply the basic calculus moves to a Hadamard and its own adjoint. Furthermore, note that our usual requirement for the Hadamards to be self-transpose is not necessary here, since we never rotate the crossings.

## (d) The Shor code

*Overview*. We present a $[[n^2, 1, n]]_d^{U(d)}$ code: that is, a code which uses $n^2$ physical qudits to encode 1 logical qudit in a way that corrects $\lfloor (n-1)/2 \rfloor$ arbitrary physical qudit errors. The data are a family of $n$ $d$-dimensional Hadamard matrices.

*Circuit*. We choose the encoding map from figure 21b.

*Specification*. Satisfaction of the conditions of theorem 6.2.

*Verification*. In figure 23, we illustrate one error configuration for the $n = 3$ case, where $e$ encodes two full qudit errors. All other cases work similarly. The general principle is the same as for §6c.

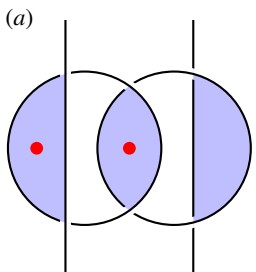 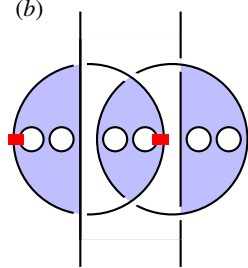

**Figure 24.** The Knill–Laflamme condition for UEB codes. (*a*) Phase code and (*b*) full qudit code. (Online version in colour.)

*Calculus.* This requires only the basic calculus.

*Novelty.* For $d = 2$, $n = 3$ and the qubit Fourier Hadamard, this is exactly Shor's 9-qubit code [14]. The qudit generalization for a general Hadamard is discussed in [12]. As with the phase code, our version is more general still, since each of the Hadamards can be different.

### (e) Unitary error basis codes

*Overview.* We show that the phase and Shor codes described above still work correctly when the Hadamards are replaced by (*UEBs*). These new codes have the same types $[[n, 1, n]]_{d^2}^{\mathcal{P}}$, $[[n^2, 1, n]]_{d^2}^{U(d)}$ as the phase and Shor codes, except with the additional restriction that the systems are of square dimension, since UEBs always have a square number of elements.

*Unitary error bases.* UEBs are fundamental structures in quantum information which play a central role in quantum teleportation and dense coding [16], and also in error correction when they satisfy the additional axioms of a *nice error basis* [17]. However, the new UEB codes we present here are seemingly unrelated, and do *not* require the additional nice error basis axioms. UEBs are defined as follows.

**Definition 6.3.** On a finite-dimensional Hilbert space $H$, a *UEB* is a basis of unitary operators $U_i : H \to H$ such that $\mathrm{Tr}(U_i^\dagger U_j) = \delta_{ij} \dim(H)$.

UEBs have an elegant presentation in terms of the graphical calculus of **2Hilb** [36,38]. Consider a vertex of the following type, where the wires with unshaded regions on both sides represent a finite-dimensional Hilbert space $H$, and the shaded region is labelled by a finite set $I$:

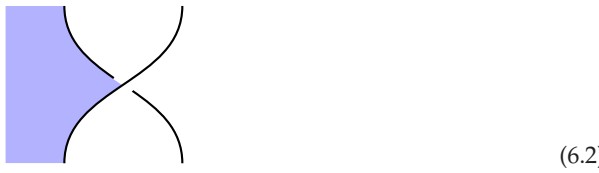

$$(6.2)$$

As per §2a, such a vertex corresponds to a family of linear maps $\{U_i : H \to H\}_{i \in I}$.

**Theorem 6.4 ([36, proposition 9]).** *A vertex of type* (6.2) *satisfies equations analogous to figure 6b and c if and only if the corresponding family of linear maps* $\{U_i : H \to H\}_{i \in I}$ *is a UEB.*

For a precise statement and proof of this theorem, see [36, proposition 9].

*Circuit*. We choose the following encoding maps to generalize the phase and Shor codes, respectively:

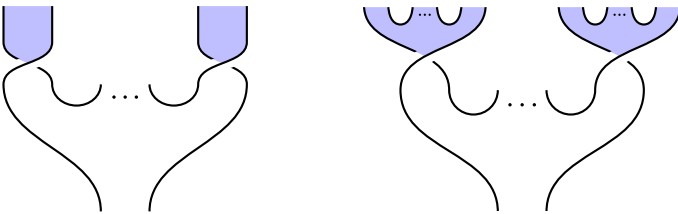

*Specification*. Satisfaction of the conditions of theorem 6.2.

*Verification*. The procedure is identical to the phase and Shor code verifications, the only difference being that some regions are differently shaded. To make this clear, in figure 24, we give the graphical representations of the Knill–Laflamme $i^\dagger \circ e \circ i$ composites for these new codes; compare these images to the first graphics in figures 22 and 23.

*Novelty*. As error correcting codes, these have precisely the same strength as the traditional phase and Shor codes. However, they are constructed from completely different data,[10] and therefore push the theory of quantum error correcting codes in a new direction. This showcases the power of our approach to uncover new paradigms in quantum information.

Data accessibility. This article does not contain any additional data.

Competing interests. We declare we have no competing interests.

Funding. J.V. acknowledges funding from the Royal Society.

Acknowledgements. We thank Paul-André Melliès for suggesting the shaded tangle representation, Amar Hadzihasanovic for detailed conversations about the protocol in §5d, Matty Hoban, Nathan Bowler and Niel de Beaudrap for telling us some useful things about cluster states, and Arthur Jaffe, Zhengwei Liu and Alex Wozniakowski for discussions about planar para algebras.

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
