## [Reviewer comments · Proceedings. Mathematical, Physical, and Engineering Sciences]

Review History

RSPA-2018-0338.R0 (Original submission)

Review form: Referee 1

Is the manuscript an original and important contribution to its field?

Yes

Is the paper of sufficient general interest?

Yes

Is the overall quality of the paper suitable?

Yes

Quality of the paper

An excellent paper making an important contribution to the field: should be published.

Can the paper be shortened without overall detriment to the main message?

No

Do you think some of the material would be more appropriate as an electronic appendix?

No

For papers with colour figures – is colour essential?

Yes

If there is supplementary material, is this adequate and clear?

Not applicable

Are there details of how to obtain materials and data, including any restrictions that may apply?

Not applicable

Do you have any ethical concerns with this paper?

No

Recommendation?

Accept with minor revision (please list in comments)

Comments to the Author(s)

This is an excellent paper, and I am very excited to read it.

It is also very nice to see the many connections to the mathematical literature.

I do appreciate the lack of formalities, and reliance on examples, but this does have drawbacks as it becomes frustrating to figure out exactly what rules are being used where.

Perhaps more notation should be introduced to label exactly which calculi is being referred to at each point in the paper?

To be completely low-brow, the point of the paper seems to be that Hadamard and control-Z operations behave like knots, and so we can reason about these using knot diagrams.

But it is not until page 10 that we find this out. I think you could make this more explicit in the introductory paragraphs. The fancy mathematics is great, but some low-brow is good too.

I do object to the use of the word "program" in the title and elsewhere. These "programs" should be called "circuits" or "operations", or even "protocols" (as is used in a few places.)

Clearly one can use the "program" terminology to refer to almost anything.

For example " $1+2$ " is a program. However, this seems to be just fancy language for something much simpler. I would reserve the "program" terminology to refer to something with "while loops", and perhaps a denotational semantics, etc. This is not a strong objection on my part, but I make it nonetheless. I also feel that calling this something like "circuit verification" would encourage a larger audience to read the work.

Many references are made to "a self-transpose Hadamard matrix", but this terminology is not defined anywhere. Several examples are given, for example a "fourier-hadamard" matrix, but what exactly is meant by "a self-transpose Hadamard matrix" in general? The terminology of "self-transpose" is not even entirely standard.

Figure 1 (a) the caption should read "Qudit identity gate".

Please provide a paragraph summarizing the relation between the following:

"The generalized graphical calculus."

Versus

"the graphical calculus"

versus

"The shaded tangle calculus"

versus

"the basic calculus"

versus

"the extended calculus".

Is a vertex in the generalized calculus supposed to correspond to (permit) a crossing in the tangle calculus?

What are the "red errors" (p21) representing in these calculi? Are these vertices? Or arbitrary unknown sub-diagrams?

p8. "Restricted calculus. We use a highly restricted part of this calculus"
Where do you use this?

p10. "Specifications. We can write our specifications using the entire language"
Which "entire language"?

Quote: "program language illustrated in Figure 1", once again you refer to one of the many languages being used in this paper without saying which one.

Line 12: "Applying our graphical calculus", yes
which one?

Proposition 3.1.

"This arises directly from the representation of GHZ states in the CQM programme [16]." Ok, but please explain how we get GHZ states using the rules in section 2.1.

Or at least confirm these rules are enough for this.

Review form: Referee 2

Is the manuscript an original and important contribution to its field?

Yes

Is the paper of sufficient general interest?

No

Is the overall quality of the paper suitable?

No

Quality of the paper

A paper that is of insufficient interest, quality or importance.

Can the paper be shortened without overall detriment to the main message?

No

Do you think some of the material would be more appropriate as an electronic appendix?

No

For papers with colour figures – is colour essential?

Yes

If there is supplementary material, is this adequate and clear?

Not applicable

Are there details of how to obtain materials and data, including any restrictions that may apply?

Not applicable

Do you have any ethical concerns with this paper?

No

Recommendation?

Reject – article is not of sufficient interest (we will consider a transfer to another journal)

Comments to the Author(s)

The paper presents a novel graphical calculus for quantum programmes and for quantum error correction. The calculus is based on diagrams of (virtual?) tangles, with additional data in the form of a shading. In this language, the authors are able to show that several known programmes apply to more general objects than those for which they were originally formulated- qudits rather than just qubits, arbitrary Hadamards rather than a more restricted class, and so on.

The description of the formalism is insufficient (detailed comments are given in the attached text file). Graphical calculi for quantum programs are a mini-industry, and there are many papers and many competing approaches. It would be useful to see some of these compared side-by-side. The applications to quantum information seem to be fairly mild generalizations that could equally well be obtained without the graphical formalism (whose local moves are, after all, a pictorial encoding of several algebraic identities- a fact the authors neglect to mention).

I do think that the paper is sound and that the results are correct and even quite nice, particularly in Sections 5 and 6.

Review form: Referee 3 (Avishy Carmi)

Is the manuscript an original and important contribution to its field?

Yes

Is the paper of sufficient general interest?

Yes

Is the overall quality of the paper suitable?

Yes

Quality of the paper

A good paper worth publishing in Proceedings.

Can the paper be shortened without overall detriment to the main message?

Yes

Do you think some of the material would be more appropriate as an electronic appendix?

Yes

For papers with colour figures – is colour essential?

Yes

If there is supplementary material, is this adequate and clear?

Not applicable

Are there details of how to obtain materials and data, including any restrictions that may apply?

Not applicable

Do you have any ethical concerns with this paper?

No

Recommendation?

Major revision is needed (please make suggestions in comments)

Comments to the Author(s)

The paper introduces a diagrammatic language whose primitives are shaded tangles for representing quantum programs together with their outputs. Although shaded diagrams are well known in representation theory, here they are used in what seems to be a novel way. Another novel aspect of the formalism is that tangle equivalence is interpreted as a verification procedure of the underlying program.

The paper is comprehensive and includes rendition of many known algorithms in quantum information theory using the proposed diagrammatic language. In an attempt to rebut the conclusions of reference [1], the authors demonstrate, in addition, some new algorithms developed entirely using shaded tangles.

Overall, I like the paper and its message. Nevertheless, my feeling is that it may not be easily comprehended by those people who are not familiar with the categorial formation of quantum mechanics, planar algebras, and low-dimensional topology.

There are long introductory parts in the beginning and throughout the paper (Novelty sections), stressing the distinctions and drawing parallels between the present formalism and existing frameworks. These are too lengthy in my taste, I virtually had to get to Section 4.6 for the interesting new parts. My suggestion to the authors is to trim these parts, potentially moving some of them to an appendix or to the last part of the paper. I believe this will improve the readability of the manuscript.

Comments (and questions):

1. Section 1.4: The authors mention their formalism is incomplete in the sense that there are quantum algorithms that cannot be represented by shaded tangles. Is there any relation here to syntactical completeness in logic? Would completeness here will result in in verification inconsistencies?

2. Section 4.6: A difficult quantum state transfer operation turns out to be a series of Reidemeister

moves in this formalism. Yet, the Overview part remains unclear about its actual implementation. Can you be more specific ?

3. Section 5.3: That robustness in the teleportation process is achieved by the extended calculus has a meaning which the authors fail to mention. Why do we need the extended calculus to achieve robustness here ? Can you be more specific on the roles of moves in Figure 5(e) and 5(f) in this respect ?

4. Section 6: this section presents some novel results and, in my view, is the most interesting. As some of the drawings are intuitive, i.e., capturing the errors by bubbles, i would suggest showing an example of an error correction scheme already in the beginning of the paper. This will motivate readers to keep on reading in an attempt to understand the graphical language. But before that could happen, I believe this section should be made consistent with rest of the paper. Using red disks and bars are somewhat hard to follow. In this respect, a better explanation to the drawing above Section 6.1 should be provided.

5. There is a similarity between the symbolic (surgical) operation of removing errors and the concept of topological (noise) resilience in:

<https://arxiv.org/abs/1409.5505>

where “bubbles” are used to represent uncontaminated parts of an information fusion (tangle) network. I leave it up to the authors to decide whether this analogy is illuminating in the context of their work.

6. Section 6.3: This section relies on a result which has been presented elsewhere. I do not immediately see how the drawing in (9) captures the notion of UEB. A proof (in the appendix) or at least some words about this result could surely help.

Review form: Referee 4 (Louis Kauffman)

Is the manuscript an original and important contribution to its field?

Yes

Is the paper of sufficient general interest?

Yes

Is the overall quality of the paper suitable?

Yes

Quality of the paper

An excellent paper making an important contribution to the field: should be published.

Can the paper be shortened without overall detriment to the main message?

No

Do you think some of the material would be more appropriate as an electronic appendix?

No

For papers with colour figures - is colour essential?

Yes

If there is supplementary material, is this adequate and clear?

Not applicable

Are there details of how to obtain materials and data, including any restrictions that may apply?

Yes

Do you have any ethical concerns with this paper?

No

Recommendation?

Accept as is

Comments to the Author(s)

This is a very well formulated 2-categorical approach to quantum information. I highly recommend its publication in its present form.

It would be helpful if the authors could discuss the possible physical meaning of the apparently topological patterns that occur in their formalism.

I mean this in analogy with the appearance of the Yang-Baxter Equation in topological quantum computing, where one sees this as a possible representation of braiding at the physical level.

Decision letter (RSPA-2018-0338.R0)

03-Jan-2019

Dear Mr Reutter

The Editor of Proceedings A has now received comments from referees on the above paper and would like you to revise it in accordance with their suggestions which can be found below (not including confidential reports to the Editor).

Please submit a copy of your revised paper within four weeks - if we do not hear from you within this time then it will be assumed that the paper has been withdrawn. In exceptional circumstances, extensions may be possible if agreed with the Editorial Office in advance.

Please note that it is the editorial policy of Proceedings A to offer authors one round of revision in which to address changes requested by referees. If the revisions are not considered satisfactory by the Editor, then the paper will be rejected, and not considered further for publication by the journal. In the event that the author chooses not to address a referee's comments, and no scientific justification is included in their cover letter for this omission, it is at the discretion of the Editor whether to continue considering the manuscript.

- Acknowledgements
- Funding statement

To revise your manuscript, log into <http://mc.manuscriptcentral.com/prsa> and enter your

Author Centre, where you will find your manuscript title listed under "Manuscripts with Decisions." Under "Actions," click on "Create a Revision." Your manuscript number has been appended to denote a revision.

You will be unable to make your revisions on the originally submitted version of the manuscript. Instead, revise your manuscript and upload a new version through your Author Centre.

When submitting your revised manuscript, you will be able to respond to the comments made by the referee(s) and upload a file "Response to Referees" in "Section 6 - File Upload". Please use this to document how you have responded to the comments, and the adjustments you have made. In order to expedite the processing of the revised manuscript, please be as specific as possible in your response to the referee(s).

IMPORTANT: Your original files are available to you when you upload your revised manuscript. Please delete any unnecessary previous files before uploading your revised version.

When revising your paper please ensure that it remains under 28 pages long. In addition, any pages over 20 will be subject to a charge (£150 + VAT (where applicable) per page). Your paper has been ESTIMATED to be 27 pages.

Once again, thank you for submitting your manuscript to Proc. R. Soc. A and I look forward to receiving your revision. If you have any questions at all, please do not hesitate to get in touch.

Yours sincerely

Alice Power
Publishing Editor
Proceedings A
proceedingsa@royalsociety.org

Reviewer(s)' Comments to Author:

Referee: 1

Comments to the Author(s)

This is an excellent paper, and I am very excited to read it.

It is also very nice to see the many connections to the mathematical literature.

I do appreciate the lack of formalities, and reliance on examples, but this does have drawbacks as it becomes frustrating to figure out exactly what rules are being used where.

Perhaps more notation should be introduced to label exactly which calculi is being referred to at each point in the paper?

To be completely low-brow, the point of the paper seems to be that Hadamard and control-Z operations behave like knots, and so we can reason about these using knot diagrams. But it is not until page 10 that we find this out. I think you could make this more explicit in the introductory paragraphs. The fancy mathematics is great, but some low-brow is good too.

I do object to the use of the word "program" in the title and elsewhere. These "programs" should be called "circuits" or "operations", or even "protocols" (as is used in a few places.) Clearly one can use the "program" terminology to refer to almost anything. For example "1+2" is a program. However, this seems to be just fancy language for something much simpler. I would reserve the "program" terminology to refer to something with "while loops", and perhaps a denotational semantics, etc. This is not a strong objection on my part, but I make it nonetheless. I also feel that calling this something like "circuit verification" would encourage a larger audience to read the work.

Many references are made to "a self-transpose Hadamard matrix", but this terminology is not defined anywhere. Several examples are given, for example a "fourier-hadamard" matrix, but what exactly is meant by "a self-transpose Hadamard matrix" in general? The terminology of "self-transpose" is not even entirely standard.

Figure 1 (a) the caption should read "Qudit identity gate".

Please provide a paragraph summarizing the relation between the following:

"The generalized graphical calculus."

Versus

"the graphical calculus"

versus

"The shaded tangle calculus"

versus

"the basic calculus"

versus

"the extended calculus".

Is a vertex in the generalized calculus supposed to correspond to (permit) a crossing in the tangle calculus?

What are the "red errors" (p21) representing in these calculi? Are these vertices? Or arbitrary unknown sub-diagrams?

p8. "Restricted calculus. We use a highly restricted part of this calculus"
Where do you use this?

p10. "Specifications. We can write our specifications using the entire language"
Which "entire language"?

Quote: "program language illustrated in Figure 1", once again you refer to one of the many languages being used in this paper without saying which one.

Line 12: "Applying our graphical calculus", yes
which one?

Proposition 3.1.

"This arises directly from the representation of GHZ states in the CQM programme [16]." Ok, but please explain how we get GHZ states using the rules in section 2.1.

Or at least confirm these rules are enough for this.

Referee: 2

Comments to the Author(s)

The paper presents a novel graphical calculus for quantum programmes and for quantum error correction. The calculus is based on diagrams of (virtual?) tangles, with additional data in the form of a shading. In this language, the authors are able to show that several known programmes apply to more general objects than those for which they were originally formulated- qudits rather than just qubits, arbitrary Hadamards rather than a more restricted class, and so on.

The description of the formalism is insufficient (detailed comments are given in the attached text file). Graphical calculi for quantum programs are a mini-industry, and there are many papers and many competing approaches. It would be useful to see some of these compared side-by-side. The applications to quantum information seem to be fairly mild generalizations that could equally well be obtained without the graphical formalism (whose local moves are, after all, a pictorial encoding of several algebraic identities- a fact the authors neglect to mention).

I do think that the paper is sound and that the results are correct and even quite nice, particularly in Sections 5 and 6.

Referee: 3

Comments to the Author(s)

The paper introduces a diagrammatic language whose primitives are shaded tangles for representing quantum programs together with their outputs. Although shaded diagrams are well known in representation theory, here they are used in what seems to be a novel way. Another novel aspect of the formalism is that tangle equivalence is interpreted as a verification procedure of the underlying program.

The paper is comprehensive and includes rendition of many known algorithms in quantum information theory using the proposed diagrammatic language. In an attempt to rebut the conclusions of reference [1], the authors demonstrate, in addition, some new algorithms developed entirely using shaded tangles.

Overall, I like the paper and its message. Nevertheless, my feeling is that it may not be easily comprehended by those people who are not familiar with the categorial formation of quantum mechanics, planar algebras, and low-dimensional topology.

There are long introductory parts in the beginning and throughout the paper (Novelty sections), stressing the distinctions and drawing parallels between the present formalism and existing frameworks. These are too lengthy in my taste, I virtually had to get to Section 4.6 for the interesting new parts. My suggestion to the authors is to trim these parts, potentially moving some of them to an appendix or to the last part of the paper. I believe this will improve the readability of the manuscript.

Comments (and questions):

1. Section 1.4: The authors mention their formalism is incomplete in the sense that there are quantum algorithms that cannot be represented by shaded tangles. Is there any relation here to syntactical completeness in logic? Would completeness here will result in in verification inconsistencies?

2. Section 4.6: A difficult quantum state transfer operation turns out to be a series of Reidemeister moves in this formalism. Yet, the Overview part remains unclear about its actual implementation. Can you be more specific ?

3. Section 5.3: That robustness in the teleportation process is achieved by the extended calculus has a meaning which the authors fail to mention. Why do we need the extended calculus to achieve robustness here ? Can you be more specific on the roles of moves in Figure 5(e) and 5(f) in this respect ?

4. Section 6: this section presents some novel results and, in my view, is the most interesting. As some of the drawings are intuitive, i.e., capturing the errors by bubbles, i would suggest showing an example of an error correction scheme already in the beginning of the paper. This will motivate readers to keep on reading in an attempt to understand the graphical language. But before that could happen, I believe this section should be made consistent with rest of the paper. Using red disks and bars are somewhat hard to follow. In this respect, a better explanation to the drawing above Section 6.1 should be provided.

5. There is a similarity between the symbolic (surgical) operation of removing errors and the concept of topological (noise) resilience in:

<https://arxiv.org/abs/1409.5505>

where “bubbles” are used to represent uncontaminated parts of an information fusion (tangle) network. I leave it up to the authors to decide whether this analogy is illuminating in the context of their work.

6. Section 6.3: This section relies on a result which has been presented elsewhere. I do not immediately see how the drawing in (9) captures the notion of UEB. A proof (in the appendix) or at least some words about this result could surely help.

Referee: 4

Comments to the Author(s)

This is a very well formulated 2-categorical approach to quantum information. I highly recommend its publication in its present form.

It would be helpful if the authors could discuss the possible physical meaning of the apparently topological patterns that occur in their formalism.

I mean this in analogy with the appearance of the Yang-Baxter Equation in topological quantum computing, where one sees this as a possible representation of braiding at the physical level.

** The application was unable to attach manuscript files to this email, because one or more of the files exceeded the allowable attachment size (6MB). **

Author's Response to Decision Letter for (RSPA-2018-0338.R0)

See Appendix A.

RSPA-2018-0338.R1 (Revision)

Review form: Referee 1

Is the manuscript an original and important contribution to its field?

Yes

Is the paper of sufficient general interest?

Yes

Is the overall quality of the paper suitable?

Yes

Quality of the paper

An excellent paper making an important contribution to the field: should be published.

Can the paper be shortened without overall detriment to the main message?

No

Do you think some of the material would be more appropriate as an electronic appendix?

No

For papers with colour figures - is colour essential?

No

If there is supplementary material, is this adequate and clear?

Yes

Are there details of how to obtain materials and data, including any restrictions that may apply?

Yes

Do you have any ethical concerns with this paper?

No

Recommendation?

Accept as is

Comments to the Author(s)

No further comment.

Review form: Referee 2

Is the manuscript an original and important contribution to its field?

Yes

Is the paper of sufficient general interest?

No

Is the overall quality of the paper suitable?

Yes

Quality of the paper

A paper that is of insufficient interest, quality or importance.

Can the paper be shortened without overall detriment to the main message?

No

Do you think some of the material would be more appropriate as an electronic appendix?

No

For papers with colour figures – is colour essential?

Yes

If there is supplementary material, is this adequate and clear?

Not applicable

Are there details of how to obtain materials and data, including any restrictions that may apply?

Not applicable

Do you have any ethical concerns with this paper?

No

Recommendation?

Reject – article is not of sufficient interest (we will consider a transfer to another journal)

Comments to the Author(s)

The paper is much improved, and criticisms have been substantively addressed.

The reviewer remains unconvinced of the general interest of this calculus; however this is the only remaining major objection.

Review form: Referee 3 (Avishy Carmi)

Is the manuscript an original and important contribution to its field?

Yes

Is the paper of sufficient general interest?

Yes

Is the overall quality of the paper suitable?

Yes

Quality of the paper

A good paper worth publishing in Proceedings.

Can the paper be shortened without overall detriment to the main message?

No

Do you think some of the material would be more appropriate as an electronic appendix?

No

For papers with colour figures – is colour essential?

Yes

If there is supplementary material, is this adequate and clear?

Not applicable

Are there details of how to obtain materials and data, including any restrictions that may apply?

Not applicable

Do you have any ethical concerns with this paper?

No

Recommendation?

Accept as is

Comments to the Author(s)

The authors have successfully addressed my concerns. I therefore recommend publication of the manuscript in its present form.

Decision letter (RSPA-2018-0338.R1)

Dear Mr Reutter

On behalf of the Editor, I am pleased to inform you that your manuscript entitled "Shaded tangles for the design and verification of quantum circuits" has been accepted in its final form for publication in Proceedings A.

Our Production Office will be in contact with you in due course. You can expect to receive a proof of your article soon. Please contact the office to let us know if you are likely to be away from e-mail in the near future. If you do not notify us and comments are not received within 5 days of sending the proof, we may publish the paper as it stands.

Open access

You are invited to opt for open access, our author pays publishing model. Payment of open access fees will enable your article to be made freely available via the Royal Society website as soon as it is ready for publication. For more information about open access please visit http://royalsocietypublishing.org/site/authors/open_access.xhtml. The open access fee for this journal is £1700/\$2380/€2040 per article. VAT will be charged where applicable.

Note that if you have opted for open access then payment will be required before the article is published – payment instructions will follow shortly. If you wish to opt for open access then please inform the editorial office (proceedingsa@royalsociety.org) as soon as possible.

Your article has been estimated as being 27 pages long. Our Production Office will inform you of the exact length at the proof stage.

Proceedings A levies charges for articles which exceed 20 printed pages. (based upon approximately 540 words or 2 figures per page). Articles exceeding this limit will incur page charges of £150 per page or part page, plus VAT (where applicable).

Under the terms of our licence to publish you may post the author generated postprint (ie. your accepted version not the final typeset version) of your manuscript at any time and this can be made freely available. Postprints can be deposited on a personal or institutional website, or a recognised server/repository. Please note however, that the reporting of postprints is subject to a media embargo, and that the status the manuscript should be made clear. Upon publication of the definitive version on the publisher's site, full details and a link should be added.

You can cite the article in advance of publication using its DOI. The DOI will take the form: 10.1098/rspa.XXXX.YYYY, where XXXX and YYYY are the last 8 digits of your manuscript number (eg. if your manuscript number is RSPA-2017-1234 the DOI would be 10.1098/rspa.2017.1234).

For tips on promoting your accepted paper see our blog post:
<https://blogs.royalsociety.org/publishing/promoting-your-latest-paper-and-tracking-your-results/>

Thank you for your submission. On behalf of the Editors of the journal, we look forward to your continued contributions to the Journal.

Best wishes

Alice Power
Publishing Editor
Proceedings A Editorial Office
proceedingsa@royalsociety.org

Reviewer(s)' Comments to Author:

Referee: 1

Comments to the Author(s)
No further comment.

Referee: 3

Comments to the Author(s)
The authors have successfully addressed my concerns. I therefore recommend publication of the manuscript in its present form.

Referee: 2

Comments to the Author(s)
The paper is much improved, and criticisms have been substantively addressed.

The reviewer remains unconvinced of the general interest of this calculus; however this is the only remaining major objection.

Appendix A

Response to Referees

We are grateful to the reviewers for their favorable reviews, and their remarkably detailed and helpful comments. Reviewer 1 writes “This is an excellent paper, and I am very excited to read it”; reviewer 2 writes “the paper is sound and that the results are correct and even quite nice”; reviewer 3 writes “Overall, I like the paper and its message”, and reviewer 4 writes “This is a very well formulated 2-categorical approach to quantum information.”

We appreciate the obvious care they have taken to make a number of important scientific points, as well as to identify typographical, stylistic and expositional errors in the manuscript. In light of these, we have updated our manuscript comprehensively, and we feel it is much improved as a result. We address the reviewers specific comments below.

We hope that our manuscript will be acceptable for publication in this revised form, and we look forward to hearing from you.

Response to Referee 1

This is an excellent paper, and I am very excited to read it. It is also very nice to see the many connections to the mathematical literature.

I do appreciate the lack of formalities, and reliance on examples, but this does have drawbacks as it becomes frustrating to figure out exactly what rules are being used where. Perhaps more notation should be introduced to label exactly which calculi is being referred to at each point in the paper?

We have significantly simplified our terminology, now referring just to the 'basic calculus' and 'extended calculus' in a clearer way. In particular, this is now much clearer in Section 2.2 where the calculus is formally defined, and is also made clearer where each protocol is described, in a paragraph headed "Calculus".

To be completely low-brow, the point of the paper seems to be that Hadamard and control-Z operations behave like knots, and so we can reason about these using knot diagrams. But it is not until page 10 that we find this out. I think you could make this more explicit in the introductory paragraphs. The fancy mathematics is great, but some low-brow is good too.

Thank you for this suggestion. We now explicitly mention Hadamard and control-Z gates in the introduction.

I do object to the use of the word "program" in the title and elsewhere. These "programs" should be called "circuits" or "operations", or even "protocols" (as is used in a few places.) Clearly one can use the "program" terminology to refer to almost anything. For example "1+2" is a program. However, this seems to be just fancy language for something much simpler. I would reserve the "program" terminology to refer to something with "while loops", and perhaps a denotational semantics, etc. This is not a strong objection on my part, but I make it nonetheless. I also feel that calling this something like "circuit verification" would encourage a larger audience to read the work.

Throughout the paper, we changed 'program' to 'circuit' or 'procedure'.

Many references are made to "a self-transpose Hadamard matrix", but this terminology is not defined anywhere. Several examples are given, for example a "fourier-hadamard" matrix, but what exactly is meant by "a self-transpose Hadamard matrix" in general? The terminology of "self-transpose" is not even entirely standard.

We added a definition of 'Hadamard matrix', and 'self-transpose' Hadamard matrix immediately before the theorem first mentioning self-transpose Hadamard matrices.

Figure 1 (a) the caption should read "Qudit identity gate".

We changed this as suggested.

Please provide a paragraph summarizing the relation between the following: "The generalized graphical calculus." Versus "the graphical calculus" versus "The shaded tangle calculus" versus "the basic calculus" versus "the extended calculus".

We have significantly reworded Section 2 to simplify our terminology. We now refer simply to the 'basic calculus' and the 'extended calculus', which are directly compared when they are introduced. All of this is a special case of the standard graphical notation for 2Hilb , the 2-category of 2-Hilbert spaces.

Is a vertex in the generalized calculus supposed to correspond to (permit) a crossing in the tangle calculus?

This is correct and is now clarified in the second paragraph of section 2.2. Indeed, every graphical component of our calculus, such as the cups/caps, tangles and the red error dots and error bars are vertices in the graphical calculus of 2Hilb .

What are the "red errors" (p21) representing in these calculi? Are these vertices? Or arbitrary unknown sub-diagrams?

They are generic vertices in 2Hilb of the given type, which make no assumptions about the form of the error which is acting locally. We have reworded the paragraph introducing the red errors, to make this clearer.

p8. "Restricted calculus. We use a highly restricted part of this calculus" Where do you use this?

We added a comment that we use this restricted part of 2Hilb throughout the whole paper.

p10. "Specifications. We can write our specifications using the entire language" Which "entire language"? Quote: "program language illustrated in Figure 1", once again you refer to one of the many languages being used in this paper without saying which one. Line 12: "Applying our graphical calculus", yes which one?

We completely reworded this paragraph, avoiding reference to any 'language' and using more descriptive terms.

Proposition 3.1. "This arises directly from the representation of GHZ states in the CQM programme [16]." Ok, but please explain how we get GHZ states using the rules in section 2.1. Or at least confirm these rules are enough for this.

We added a sentence explaining that this follows from composing the expressions of the cups in Section 2.1. according to the general rules of the graphical calculus of 2Hilb .

Lastly, we would like to thank the reviewer for their careful reading of the manuscript, and their many useful comments.

Response to Referee 2

The paper presents a novel graphical calculus for quantum programmes and for quantum error correction. The calculus is based on diagrams of (virtual?) tangles, with additional data in the form of a shading. In this language, the authors are able to show that several known programmes apply to more general objects than those for which they were originally formulated- qudits rather than just qubits, arbitrary Hadamards rather than a more restricted class, and so on.

Graphical calculi for quantum programs are a mini-industry, and there are many papers and many competing approaches. It would be useful to see some of these compared side-by-side. The applications to quantum information seem to be fairly mild generalizations that could equally well be obtained without the graphical formalism (whose local moves are, after all, a pictorial encoding of several algebraic identities- a fact the authors neglect to mention).

Thank you for this comment. In Section 1.5, the revised document now contains references to a wide range of diagrammatic approaches to quantum information, including the ZX calculus, para planar algebras, the tangle machines of Carmi and Moskvovich, work of Kauffman, and also topological quantum computation. We feel that a full survey-like section comparing all such diagrammatic languages would be out of place here, although such a survey would of course be nice to read as an independent document. If the reviewer feels that we have missed out any key related work, we are happy to expand this section with additional citations and discussion.

I do think that the paper is sound and that the results are correct and even quite nice, particularly in Sections 5 and 6.

The description of the formalism is insufficient (detailed comments are given in the attached text file) [copied inline here].

Abstract:

- 1) What does "isotopic" mean? Either "ambient isotopic" (in what ambient space?) or "equivalent".*
- 2) What are "equivalent programs"?*
- 3) What does "many" mean?*
- 4) What makes the insight substantial? Could you give an example of a "substantial insight"?*
- 5) Last sentence is too vague.*

Thank you for these comments on the abstract. We have thoroughly revised it to satisfy these points.

Section 1.1 is weak. Paragraph 1: "look like traditional knot diagrams except": It seems to me that the major difference is the height function.

We have modified the second paragraph of Section 1.1 to make this clearer.

"decorated with a shading pattern": vague

We have made this more precise in the first paragraph of Section 1.1.

Paragraph 2: "an operational semantics"- what does this mean?

This is terminology from theoretical computer science, meaning a mathematical model. We have reworded this to use clearer terminology.

Paragraph 3: First sentence is unclear. So is the second- what are "equal interpretations"?

We have reworded this paragraph to make it clearer and more precise.

Paragraph 4: "powerful" - what makes it powerful?

"humans have an innate skill for visualizing knot isotopy"- do we? Can we visualize the ambient isotopy between the Perko pair?

We have changed the phrasing in this paragraph to address these reasonable points. We of course agree that the Perko pair isotopy is not easy to visualize.

Section 1.2

Paragraph 1: "phenomena"- su Is the formalism of shaded tangles novel in the context of quantum computation? I was left without understanding whether the paper generalizes some results from qubits to qudits and maybe presents a few more algorithms, or whether there is more to it.

Thank you for this thoughtful response. Although our approach is novel, we certainly make no claim that this is the first application of tangles to quantum information. This is made clear by Section 1.5 on related work, and we have now added a forward reference to this.

In this part of Section 1.2, we highlight the novel aspects of the quantum procedures that we describe. Some of these are generalized, and some of these are completely new. Science proceeds iteratively, and we do not feel it is right to make a claim as to whether there is 'more to it'. We feel that these novel aspects which we highlight here are strong in themselves, and suitable for publication in their present form.

Insight: I was unconvinced by this argument. Visualization within a formalism is not insight.

We have reworded and weakened this statement. For us it has the quality of 'insight', since we used it to produce the new knowledge---the nontrivial generalized and new procedures that we describe in the paper---while at the same time giving us a subjective feeling that we understand 'why' these generalized and new procedures work. But this is ultimately a subjective judgement, and the wording now appropriately reflects that.

Section 1.5 "and this paper develops these ideas further." - in what way?

We have clarified this point.

"Unlike the ZX calculus, our calculus is purely topological" - what does this mean? In what sense is this calculus more topological than the ZX calculus?

We have clarified this point. The point is that the ZX calculus also involves algebraic equalities without a direct topological interpretation.

Several other diagrammatic approaches (using tangles, virtual crossings, and suchlike) to similar problems are not mentioned, including work of Kauffman, Buliga, and Carmi-Moskovich.

Thank you for pointing out this important related work, which is now given appropriate discussion in Section 1.5.

Section 2.1. In common parlance, "equal" means "equal". Here it means something else. The word the authors are searching for is "equivalent", I believe.

Thank you for this suggestion. We have changed 'equal' to 'equivalent' as suggested.

Section 2.2. No citation is given for shaded tangle diagrams, nor is a proper definition presented, so that it is unclear to the author exactly which class of diagrams is being discussed. In particular, are there virtual crossings? (Section 5).

Our calculus is the precisely one defined by the graphical calculus of 2Hilb , induced by the vertices given as expression (2). We have added a very explicit third paragraph to Section 2.2 where we make absolutely precise the calculus that we are working with. Also, additional citations to shaded tangles are now given in Section 1.5.

Letters in circles? (Section 6).

The letter 'C' in a circle represents a controlled unitary, and is introduced in the prose in Section 5.4. The previous draft had a typo here, with this gate referred to as 'U' in its first mention; this is now corrected.

Red dots and red squares? (Section 6).

The red dots represent an arbitrary phase error, and the red squares represent an arbitrary single-qubit error. This has been made clearer where they are introduced, just before Section 6.1.

Is the reverse shaded crossing included (Equation 2) as in Figure 5f?

Yes, this is included, and we have now made this clearer in the second paragraph of Section 2.2.

Shaded diagrams, and many generalizations, have been common in representation theory for many years. This is not cited or substantially mentioned.

Thank you for pointing out this omission, which we have corrected with a reference to an excellent survey of Kauffman in the 'Tangles' paragraph of Section 1.5.

It is unclear whether the class of diagrams being discussed is novel, or coincides with a known formalism (this relates to the previous comment), except for the citation of Jones, which looks very different.

We now make a very precise statement in the third paragraph of Section 2.2 describing exactly the class of diagrams we are using. We claim no novelty for this class of shaded diagrams.

The citation [52, Section 2.4] is a horrible citation for the Reidemeister moves, particularly as these are not the Reidemeister moves for tangles.

Thank you for this point. We have replaced this with a more suitable reference to the survey of Kauffman.

As a result of this poor citation, it is unclear what Reidemeister moves even mean in this section- for example, is the transformation of a straight line to a rotated S-shape (swallowtail) considered a Reidemeister move or not?

We have restructured Section 2.2 to give a much more explicit definition of our calculus, in particular in the third paragraph of that section. This refers back to Figure 5, which shows that the "rotated S-shape" move is indeed required (although we don't refer to this as one of the Reidemeister moves.)

[39] Doesn't Jones discuss invariants of links, not of tangles?

That's correct; nonetheless, the work is closely related. We have clarified this point in the text, to make it clear that Jones' work is on links rather than tangles.

Theorem 2.1: As it is unclear to me which set of Reidemeister moves is being referred to, I cannot understand this proof (as given in the appendix). Whichever statement is meant, it is not original- a citation would be most welcome. "we can classify representations of the basic calculus as follows." - only finite dimensional representations are classified (as it says further on down)

The moves being referred to are exactly the moves of our extended calculus, which is now defined much more precisely in Section 2.2. We certainly do not claim that this idea is original; this point has been made much clearer.

Section 2.3 "all the cups and caps arising from (1) and their adjoints." - is it not the case that the caps represent the adjoints of the cups?

Thank you for this point. As suggested, we changed this to "using the cups from (1) and their adjoint caps".

"Some programs we analyze require only the basic calculus to be satisfied" - Here a principled doubt arises- the basic calculus does not conform to tangle-theoretical intuitions, as R1 and R3 moves are excluded. What is the added value of the calculus in this case? Is it just an analogue to the ZX calculus? This is probably just a matter of presentation.

The added value is simply that it is interesting to note that, for some applications, the full strength of the axiomatics is not required. This leads to a much larger class of models. We have now made this clearer in the last paragraph of Section 2.

Section 5: Here the graphical calculus suddenly changes, which is jarring. Although I can probably guess the local moves, they are not stated anywhere.

We have now added a considerable amount of extra illustration and discussion about the extra features of the graphical calculus which we are now using, including a discussion of the local moves, and an emphasis that it is precisely defined as an instance of the graphical calculus of monoidal 2-categories. The discussion in Section 2 of this monoidal structure is also now more detailed, in the third paragraph of the "Elementary description" part of Section 2.1.

Section 5.1: Insufficient citation for "shaded virtual knots". Shaded virtual knots seem not to be mentioned in the citation. It is in fact not clear to the reviewer exactly what construction is being referred to. "this overlapping is described by the monoidal structure of the 2-category." - This sentence is unclear. Which 2-category and what is monoidal? Why does it have anything to do with overlapping of 2-regions? If the 2-regions may overlap, why not just colour a normal vector to an arc or something?

We have made this much clearer, emphasizing that the formal foundation for our work is the graphical calculus of 2Hilb as a monoidal 2-category. Its graphical calculus, with overlapping marked regions, is standard, and while the reviewer's proposal of arcs equipped with coloured normal vectors could be workable, we prefer to keep the standard approach.

Section 5.2. It is not easy to see that the basic calculus performs this transformation- indeed, what is the basic calculus when there are virtual crossings present? This needs to be expanded, perhaps in an appendix.

It is not clear to the authors what the reviewer means by "this transformation". Everything remains exactly the same as before: we are still just using the basic or extended calculus, involving four types primary types of crossing (2) connected by coloured sheets, as an instance of the graphical calculus of the monoidal 2-category 2Hilb . In this graphical

calculus, sheets can in general overlap, as we explain in Section 2 where the calculus is introduced. It just so happens that in Sections 3, 4 and 6 we do not need to use this overlapping. In Section 4 we do make use of it in a minimal way, and we give extensive discussion of this part of the calculus in Section 4.1, to prepare the reader for that.

Section 5.3 "By isotopy, the entire tangle error can be pulled up, 'underneath' the lower diagonal strand, inverting its shading. The lower 'cup' of the GHZ state can then be pulled up similarly" This is unclear. A picture (or a sequence of images illustrating this sequence of transformations) is worth a thousand words.

This is a good point, we agree that this isotopy was hard to comprehend. We have addressed this with a new figure showing the key central step in the isotopy.

Section 5.4 What is "tangle gate"?

The figure which appears under the header 'Section 5.4' belongs to section 5.3. In the corresponding description of the program, we state that by 'tangle gate' we mean an arbitrary shaded tangle with the given boundary and refer back to section 3.3. where this is defined.

Section 6: Diagram 17 is not part of the calculus. What is C?

The letter 'C' in a circle is a vertex in the graphical notation of 2Hilb which represents a controlled unitary, and is introduced in the prose in Section 5.4. The previous draft had a typo here, with this gate referred to as 'U' in its first mention; this is now corrected.

Also, it appears to use a move that is "opposite" of move (c) in Figure 5.

Using Figure 5(a) and (d), the 'oppositely shaded' version of (c) is equivalent to the move (b). More generally, we show in Theorem 2.1 that in the graphical calculus of 2Hilb , every shaded isotopy follows from the moves in Figure 5.

Representing errors: This is a "new" graphical notation. Is it part of the calculus or not?

We reworded the paragraph before Section 6.1 making clearer that the red disks and bars denote arbitrary vertices of the given type in the graphical notation of 2Hilb .

Section 6.1: "A major novel feature is the visceral sense of how the protocol works that Figure 19 conveys: the phase errors are 'captured by bubbles' and turned into scalar factors. We believe this intuition has not been described elsewhere." - this is poetic, but what content is behind it? If you represent errors as circles instead of as equations, what intuition is captured? That having been said, I agree that this representation is very nice.

This is a good point; it is similar to the point made above regarding "insight". The content behind this intuition is the mathematical fact that the error correction procedure is indeed correct, which we prove here using our tangle-theoretic methods. As to why this intuition is

correct, or what deeper meaning it might have, we are not sure. We have added some words to this effect at this point in the paper.

Diagram for Theorem 6.4: Not part of the calculus.

We reworded Theorem 6.4 and the paragraph leading up to it, clarifying that the 'half-shaded' crossing is a vertex in the graphical notation of 2Hilb , fulfilling equations as specified in the theorem.

We are grateful to the reviewer for their careful reading of our manuscript, and the many positive suggestions.

Response to Referee 3

The paper introduces a diagrammatic language whose primitives are shaded tangles for representing quantum programs together with their outputs. Although shaded diagrams are well known in representation theory, here they are used in what seems to be a novel way. Another novel aspect of the formalism is that tangle equivalence is interpreted as a verification procedure of the underlying program.

The paper is comprehensive and includes rendition of many known algorithms in quantum information theory using the proposed diagrammatic language. In an attempt to rebut the conclusions of reference [1], the authors demonstrate, in addition, some new algorithms developed entirely using shaded tangles.

Overall, I like the paper and its message. Nevertheless, my feeling is that it may not be easily comprehended by those people who are not familiar with the categorial formation of quantum mechanics, planar algebras, and low-dimensional topology.

There are long introductory parts in the beginning and throughout the paper (Novelty sections), stressing the distinctions and drawing parallels between the present formalism and existing frameworks. These are too lengthy in my taste, I virtually had to get to Section 4.6 for the interesting new parts. My suggestion to the authors is to trim these parts, potentially moving some of them to an appendix or to the last part of the paper. I believe this will improve the readability of the manuscript.

This is a good point. We certainly understand that for some readers these lengthy discussions of related work and novelty may be too lengthy. However, for other readers we feel they may give critical context, and indeed other reviewers have suggested we lengthen these parts. On balance, we therefore feel it is correct to keep these parts in place. We have added a comment at the end of Section 1.1 that the reader mostly interested in the mathematical developments should skip ahead to Section 2.

Comments (and questions):

1. Section 1.4: The authors mention their formalism is incomplete in the sense that there are quantum algorithms that cannot be represented by shaded tangles. Is there any relation here to syntactical completeness in logic? Would completeness here will result in in verification inconsistencies?

Yes, we are using the standard notion of logical completeness here, for a syntax over a given model. It is not clear to us what the reviewer means by “verification inconsistencies”. Our system is sound, meaning that if we verify a property using our system, then the corresponding quantum circuit is guaranteed to have that property.

2. Section 4.6: A difficult quantum state transfer operation turns out to be a series of Reidemeister moves in this formalism. Yet, the Overview part remains unclear about its actual implementation. Can you be more specific?

Our work is at a logical level, and is independent of how the circuits may be physically implemented. We have added a footnote to this effect on page 2.

3. Section 5.3: That robustness in the teleportation process is achieved by the extended calculus has a meaning which the authors fail to mention. Why do we need the extended calculus to achieve robustness here ? Can you be more specific on the roles of moves in Figure 5(e) and 5(f) in this respect ?

We clarified that only 5(f) is needed to allow arbitrary tangle errors, which might themselves involve crossings, to move upwards.

4. Section 6: this section presents some novel results and, in my view, is the most interesting. As some of the drawings are intuitive, i.e., capturing the errors by bubbles, i would suggest showing an example of an error correction scheme already in the beginning of the paper. This will motivate readers to keep on reading in an attempt to understand the graphical language.

We now show an example verification of an error correcting code in the introduction.

But before that could happen, I believe this section should be made consistent with rest of the paper. Using red disks and bars are somewhat hard to follow. In this respect, a better explanation to the drawing above Section 6.1 should be provided.

We reworded the paragraph before Section 6.1. making clearer that the red disks and bars denote arbitrary vertices of the given type in the graphical notation of 2Hilb .

5. There is a similarity between the symbolic (surgical) operation of removing errors and the concept of topological (noise) resilience in <https://arxiv.org/abs/1409.5505>, where “bubbles” are used to represent uncontaminated parts of an information fusion (tangle) network. I leave it up to the authors to decide whether this analogy is illuminating in the context of their work.

Thank you for pointing out this interesting related work, which we now discuss in Section 1.5.

6. Section 6.3: This section relies on a result which has been presented elsewhere. I do not immediately see how the drawing in (9) captures the notion of UEB. A proof (in the appendix) or at least some words about this result could surely help.

We changed the presentation and reworded the theorem, explaining that vertices of the depicted type correspond to families of linear maps $\{U_i: H \rightarrow H\}_{i \in I}$. For the proof that such a vertex fulfills the Reidemeister 2 equations if and only if the family $\{U_i\}$ is a UEB, we refer more clearly to Proposition 9 of [Reutter, Vicary -- Biunitary constructions in quantum information].

We are grateful to the author for the careful reading of our manuscript, and the many positive suggestions.

Response to Referee 4

This is a very well formulated 2-categorical approach to quantum information. I highly recommend its publication in its present form.

It would be helpful if the authors could discuss the possible physical meaning of the apparently topological patterns that occur in their formalism.

I mean this in analogy with the appearance of the Yang-Baxter Equation in topological quantum computing, where one sees this as a possible representation of braiding at the physical level.

We added a section 'Tangles' to the related work section discussing the relation of our calculus to topological quantum computation. At the moment, we do not know whether our 'shaded braidings' have a direct physical interpretation. This is an important area to consider for future work.

We are grateful to the reviewer for these comments on our manuscript.